# UNIVERSALIZING WEAK SUPERVISION

**Changho Shin, Winfred Li, Harit Vishwakarma, Nicholas Roberts, Frederic Sala**
Department of Computer Sciences
University of Wisconsin-Madison
`{cshin23, wli525, hvishwakarma, ncroberts2, fsala}@wisc.edu`

## ABSTRACT

Weak supervision (WS) frameworks are a popular way to bypass hand-labeling large datasets for training data-hungry models. These approaches synthesize multiple noisy but cheaply-acquired estimates of labels into a set of high-quality pseudolabels for downstream training. However, the synthesis technique is specific to a particular kind of label, such as binary labels or sequences, and each new label type requires manually designing a new synthesis algorithm. Instead, we propose a universal technique that enables weak supervision over any label type while still offering desirable properties, including practical flexibility, computational efficiency, and theoretical guarantees. We apply this technique to important problems previously not tackled by WS frameworks including learning to rank, regression, and learning in hyperbolic space. Theoretically, our synthesis approach produces a consistent estimators for learning some challenging but important generalizations of the exponential family model. Experimentally, we validate our framework and show improvement over baselines in diverse settings including real-world learning-to-rank and regression problems along with learning on hyperbolic manifolds.

## 1    INTRODUCTION

Weak supervision (WS) frameworks help overcome the labeling bottleneck: the challenge of building a large dataset for use in training data-hungry deep models. WS approaches replace hand-labeling with synthesizing multiple noisy but cheap sources, called labeling functions, applied to unlabeled data. As these sources may vary in quality and be dependent, a crucial step is to model their accuracies and correlations. Informed by this model, high-quality pseudolabels are produced and used to train a downstream model. This simple yet flexible approach is highly successful in research and industry settings (Bach et al., 2019; Ré et al., 2020) .

WS frameworks offer three advantages: they are (i) flexible and subsume many existing ways to integrate side information, (ii) computationally efficient, and (iii) they offer theoretical guarantees, including estimator consistency. Unfortunately, these benefits come at the cost of being particular to very specific problem settings: categorical, usually binary, labels. Extensions, e.g., to time-series data (Safranchik et al., 2020), or to segmentation masks (Hooper et al., 2021), require a new model for source synthesis, a new algorithm, and more. We seek to side-step this expensive one-at-a time process via a *universal approach*, enabling WS to work in *any* problem setting while still providing the three advantageous properties above.

The main technical challenge for universal WS is the diversity of label settings: classification, structured prediction, regression, rankings, and more. Each of these settings seemingly demands a different approach for learning the source synthesis model, which we refer to as the *label model*. For example, Ratner et al. (2019) assumed that the distribution of the sources and latent true label is an Ising model and relied on a property of such distributions: that the inverse covariance matrix is graph-structured. It is not clear how to lift such a property to spaces of permutations or to Riemannian manifolds.

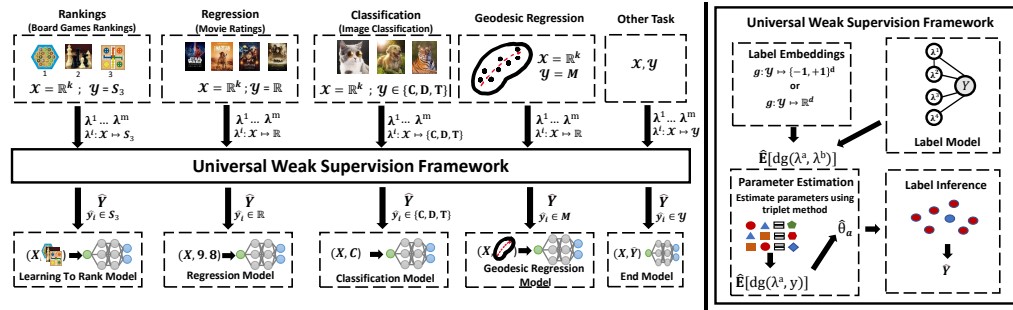

Figure 1: Applications enabled by our approach (left) and weak supervision pipeline (right).

We propose a general recipe to handle any type of label. The data generation process for the weak sources is modeled with an exponential family distribution that can represent a label from any metric space $(\mathcal{Y}, d_{\mathcal{Y}})$. We embed labels from $\mathcal{Y}$ into two tractable choices of space: the Boolean hypercube $\{\pm 1\}^d$ and Euclidean space $\mathbb{R}^d$. The label model (used for source synthesis) is learned with an efficient method-of-moments approach in the embedding space. It only requires solving a number of scalar linear or quadratic equations. Better yet, for certain cases, we show this estimator is consistent via finite sample bounds.

Experimentally, we demonstrate our approach on five choices of problems never before tackled in WS:

- **Learning rankings:** on two real-world rankings tasks, our approach with as few as five sources performs better than supervised learning with a smaller number of true labels. In contrast, an adaptation of the Snorkel (Ratner et al., 2018) framework cannot reach this performance with as many as 18 sources.
- **Regression:** on two real-world regression datasets, when using 6 or more labeling function, the performance of our approach is comparable to fully-supervised models.
- **Learning in hyperbolic spaces:** on a geodesic regression task in hyperbolic space, we consistently outperform fully-supervised learning, even when using only 3 labeling functions (LFs).
- **Estimation in generic metric spaces:** in a synthetic setting of metric spaces induced by random graphs, we demonstrate that our method handles LF heterogeneity better than the majority vote baseline.
- **Learning parse trees:** in semantic dependency parsing, we outperform strong baseline models.

In each experiment, we confirm the following desirable behaviors of weak supervision: (i) more high-quality and independent sources yield better pseudolabels, (ii) more pseudolabels can yield better performance compared to training on fewer clean labels, (iii) when source accuracy varies, our approach outperforms generalizations of majority vote.

## 2 PROBLEM FORMULATION AND LABEL MODEL

We give background on weak supervision frameworks, provide the problem formulation, describe our choice of universal label model, special cases, the embedding technique we use to reduce our problem to two easily-handled cases, and discuss end model training.

**Background:** WS frameworks are a principled way to integrate weak or noisy information to produce label estimates. These weak sources include small pieces of code expressing heuristic principles, crowdworkers, lookups in external knowledge bases, pretrained models, and many more (Karger et al., 2011; Mintz et al., 2009; Gupta & Manning, 2014; Dehghani et al., 2017; Ratner et al., 2018). Given an unlabeled dataset, users construct a set of *labeling functions* (LFs) based on weak sources and apply them to the data. The estimates produced by each LF are synthesized to produce pseudolabels that can be used to train a downstream model.

**Problem Formulation:** Let $x_1, x_2, \ldots, x_n$ be a dataset of unlabeled datapoints from $\mathcal{X}$. Associated with these are labels from an arbitrary metric space $\mathcal{Y}$ with metric $d_{\mathcal{Y}}$[1]. In conventional supervised learning, we would have pairs $(x_1, y_1), \ldots, (x_n, y_n)$; however, in WS, we *do not* have access to the labels. Instead, we have estimates of $y$ from $m$ labeling functions (LFs). Each such LF $s : \mathcal{X} \to \mathcal{Y}$ produces an estimate of the true label $y$ from a datapoint $x$. We write $\lambda^a(i) \in \mathcal{Y}$ for the output of the $a$th labeling function $s^a$ applied to the $i$th sample. Our goal is to obtain an estimate of the true label $y_i$ using the LF outputs $\lambda^1(i), \ldots, \lambda^m(i)$. This estimate $\hat{y}$, is used to train a downstream model. To produce it, we learn the *label model* $P(\lambda^1, \lambda^2, \ldots, \lambda^m | y)$. The main challenge is that *we never observe samples of $y$; it is latent*.

We can summarize the weak supervision procedure in two steps:

- **Learning the label model**: use the samples $\lambda^a(i)$ to learn the label model $P(\lambda^1, \lambda^2, \ldots, \lambda^m | y)$,
- **Perform inference**: compute $\hat{y}_i$, or $P(y_i | \lambda^1(i), \ldots, \lambda^m(i))$, or a related quantity.

**Modeling the sources** Previous approaches, e.g., Ratner et al. (2019), select a particular parametric choice to model $p(\lambda^1, \ldots, \lambda^m | y)$ that balances two goals: (i) model richness that captures differing LF accuracies and correlations, and (ii) properties that permit efficient learning. Our setting demands greater generality. However, we still wish to exploit the properties of exponential family models. The natural choice is

$$p(\lambda^1, \ldots, \lambda^m | y) = \frac{1}{Z} \exp\Big( \underbrace{\sum_{a=1}^{m} -\theta_a d_{\mathcal{Y}}(\lambda^a, y)}_{\text{Accuracy Potentials}} + \underbrace{\sum_{(a,b) \in E} -\theta_{a,b} d_{\mathcal{Y}}(\lambda^a, \lambda^b)}_{\text{Correlation Potentials}} \Big). \qquad (1)$$

Here, the set $E$ is a set of correlations corresponding to the graphical representation of the model (Figure 1, right). Observe how source quality is modeled in (1). If the value of $\theta_a$ is very large, any disagreement between the estimate $\lambda^a$ and $y$ is penalized through the distance $d_{\mathcal{Y}}(\lambda^a, y)$ and so has low probability. If $\theta_a$ is very small, such disagreements will be common; the source is *inaccurate*.

We also consider a more general version of (1). We replace $-\theta_a d_{\mathcal{Y}}(\lambda^a, y)$ with a per-source distance $d_{\theta_a}$. For example, for $\mathcal{Y} = \{\pm 1\}^d$, $d_{\theta_a}(\lambda^a, y) = -\theta_a^T |\lambda^a - y|$, with $\theta_a \in \mathbb{R}^d$, generalizes the Hamming distance. Similarly, for $\mathcal{Y} = \mathbb{R}^d$, we can generalize $-\theta_a \|\lambda^a - y\|^2$ with $d_{\theta_a}(\lambda^a, y) = -(\lambda^a - y)^T \theta_a (\lambda^a - y)$, with $\theta_a \in \mathbb{R}^{d \times d}$ p.d., so that LF errors are not necessarily isotropic. In the Appendix B, we detail variations and special cases of such models along with relationships to existing weak supervision work. Below, we give a selection of examples, noting that the last three cannot be tackled with existing methods.

- **Binary classification**: $\mathcal{Y} = \{\pm 1\}$, $d_{\mathcal{Y}}$ is the Hamming distance: this yields a shifted Ising model for standard binary classification, as in Fu et al. (2020).
- **Sequence learning**: $\mathcal{Y} = \{\pm 1\}^d$, $d_{\mathcal{Y}}$ is the Hamming distance: this yields an Ising model for sequences, as in Sala et al. (2019) and Safranchik et al. (2020).
- **Ranking**: $\mathcal{Y} = S_\rho$, the permutation group on $\{1, \ldots, \rho\}$, $d_{\mathcal{Y}}$ is the Kendall tau distance. This is a *heterogenous Mallows model*, where rankings are produced from varying-quality sources. If $m = 1$, we obtain a variant of the conventional Mallows model (Mallows, 1957).
- **Regression**: $\mathcal{Y} = \mathbb{R}$, $d_{\mathcal{Y}}$ is the squared $\ell_2$ distance: it produces sources in $\mathbb{R}^d$ with Gaussian errors.
- **Learning on Riemannian manifolds**: $\mathcal{Y} = M$, a Riemannian manifold (e.g., hyperbolic space), $d_{\mathcal{Y}}$ is the Riemannian distance $d_M$ induced by the space's Riemannian metric.

**Majority Vote (MV) and its Relatives** A simplifying approach often used as a baseline in weak supervision is the *majority vote*. Assume that the sources are conditionally independent (i.e. $E$ is empty in (1)) and all accuracies are identical. In this case, there is no need to learn the model (1); instead, the most "popular"

---

[1] We slightly abuse notation by allowing $d_{\mathcal{Y}}$ to be $d^c$, where $d$ is some base metric and $c$ is an exponent. This permits us to use, for example, the squared Euclidean distance—not itself a metric—without repeatedly writing the exponent $c$.

| Problem | Set | Distance | MV Equivalent | Im(g) | End Model |
|---------|-----|----------|---------------|-------|-----------|
| Binary Classification | $\{\pm 1\}$ | $\ell_1$ | Majority Vote | $\{\pm 1\}$ | Binary Classifier |
| Ranking | $S_\rho$ | Kendall tau | Kemeny Rule | $\{\pm 1\}^{\binom{\rho}{2}}$ | Learning to Rank |
| Regression | $\mathbb{R}$ | squared-$\ell_2$ | Arithmetic Mean | $\mathbb{R}$ | Linear Regression |
| Riemannian Manifold | $M$ | Riemannian | Fréchet Mean | $\mathbb{R}^d$ | Geodesic Regression |
| Dependency Parsing | $\mathcal{T}$ | $\ell_2$ | Fréchet Mean | $\mathbb{R}^{d \times d}$ | Parsing Model |

Table 1: A variety of problems enabled by universal WS, with specifications for sets, distances, and models.

label is used. For binary labels, this is the majority label. In the universal setting, a natural generalization is

$$\hat{y}_{\text{MV}} = \arg\min_{z \in \mathcal{Y}} \frac{1}{m} \sum\nolimits_{a=1}^{m} d_{\mathcal{Y}}(\lambda^a, z). \tag{2}$$

Special cases of (2) have their own name; for $S_\rho$, it is the Kemeny rule (Kemeny, 1959). For $d_M$, the squared Riemannian manifold distance, $\hat{y}_{\text{MV}}$ is called the *Fréchet* or *Karcher mean*.

Majority vote, however, is insufficient in cases where there is variation in the source qualities. We must learn the label model. However, generically learning (1) is an ambitious goal. Even cases that specialize (1) in multiple ways have only recently been fully solved, e.g., the permutation case for identical $\theta_a$'s was fully characterized by Mukherjee (2016). To overcome the challenge of generality, we opt for an embedding approach that reduces our problem to two tractable cases.

**Universality via Embeddings** To deal with the very high level of generality, we reduce the problem to just two metric spaces: the boolean hypercube $\{-1, +1\}^d$ and Euclidean space $\mathbb{R}^d$. To do so, let $g : \mathcal{Y} \to \{\pm 1\}^d$ (or $g : \mathcal{Y} \to \mathbb{R}^d$) be an injective *embedding function*. The advantage of this approach is that if $g$ is *isometric*—distance preserving—then probabilities are preserved under $g$. This is because the sufficient statistics are preserved: for example, for $g : \mathcal{Y} \to \mathbb{R}^d$, $-\theta_a d_{\mathcal{Y}}(\lambda^a, y) = -\theta_a d(g(\lambda^a), g(y)) = -\theta_a \|g(\lambda^a) - g(y)\|^c$, so that if $g$ is a bijection, we obtain a multivariate normal for $c = 2$. If $g$ is not isometric, there is a rich literature on *low-distortion* embeddings, with Bourgain's theorem as a cornerstone result (Bourgain, 1985). This can be used to bound the error in recovering the parameters for any label type.

**End Model** Once we have produced pseudolabels—either by applying generalized majority vote or by learning the label model and performing inference—we can use the labels to train an end model. Table 2 summarizes examples of problem types, explaining the underlying label set, the metric, the generalization of majority vote, the embedding space $\text{Im}(g)$, and an example of an end model.

## 3 UNIVERSAL LABEL MODEL LEARNING

Now that we have a specification of the label model distribution (1) (or its generalized form with per-source distances), we must learn the distribution from the observed LFs $\lambda^1, \ldots, \lambda^m$. Afterwards, we can perform inference to compute $\hat{y}$ or $p(y|\lambda^1, \ldots, \lambda^m)$ or a related quantity, and use these to train a downstream model. A simplified model with an intuitive explanation for the isotropic Gaussian case is given in Appendix D.

**Learning the Universal Label Model** Our general approach is described in Algorithm 1; its steps can be seen in the pipeline of Figure 1. It involves first computing an embedding $g(\lambda^a)$ into $\{\pm 1\}^d$ or $\mathbb{R}^d$; we use multidimensional scaling into $\mathbb{R}^d$ as our standard. Next, we learn the per-source *mean parameters*, then finally compute the *canonical parameters* $\theta_a$. The mean parameters are $\mathbb{E}[d_G(g(\lambda^a)g(y))]$; which reduce to moments like $\mathbb{E}[g(\lambda^a)_i g(y)_i]$. Here $d_G$ is some distance function associated with the embedding space. To estimate the mean parameters without observing $y$ (and thus not knowing $g(y)$), we exploit observable quantities and conditional independence. As long as we can find, for each LF, two others that are mutually conditionally independent, we can produce a simple non-linear system over the three sources (in each component of the moment). Solving this system recovers the mean parameters up to sign. We recover these as long as the LFs are better than random on average (see Ratner et al. (2019)).

---

**Algorithm 1:** Universal Label Model Learning

---

**Input:** Output of labeling functions $\lambda^a(i)$, correlation set $E$, prior $p$ for $Y$, optionally embedding function $g$.
**Embedding:** If $g$ is not given, use Multidimensional Scaling (MDS) to obtain embeddings $g(\lambda^a) \in \mathbb{R}^d \; \forall a$
**for** $a \in \{1, 2, \ldots, m\}$ **do**
    For $b : (a, b) \in E$    **Estimate Correlations:** $\forall i, j, \; \hat{\mathbb{E}}\left[g(\lambda^a)_i g(\lambda^b)_j\right] = \frac{1}{n}\sum_{t=1}^n g(\lambda^a(t))_i g(\lambda^b(t))_j$
    **Estimate Accuracy:** Pick $b, c : (a, b) \notin E, (a, c) \notin E, (b, c) \notin E$.
    **if** $\mathbf{Im}(g) = \{\pm 1\}^d$
    $\forall i$, Estimate $\hat{O}_{a,b} = \hat{P}(g(\lambda^a)_i = 1, g(\lambda^b)_i = 1)$, $\hat{O}_{a,c}, \hat{O}_{b,c}$, Estimate $\ell_a = \hat{P}(g(\lambda^a)_i = 1)$, $\ell_b, \ell_c$
        Accuracies $\leftarrow$ QUADRATICTRIPLETS$(O_{a,b}, O_{a,c}, O_{b,c}, \ell_a, \ell_b, \ell_c, p, i)$
    **else** $\forall i$, Estimate $\hat{e}_{a,b} := \hat{\mathbb{E}}\left[g(\lambda^a)_i g(\lambda^b)_i\right] = \frac{1}{n}\sum_{t=1}^n g(\lambda^a(t))_i g(\lambda^b(t))_i$, $\hat{e}_{a,c}, \hat{e}_{b,c}$
    Accuracies $\leftarrow$ CONTINUOUSTRIPLETS$(\hat{e}_{a,b}, \hat{e}_{a,c}, \hat{e}_{b,c}, p, i)$
    Recover accuracy signs (Fu et al., 2020)
**end for**
**return** $\hat{\theta}_a, \hat{\theta}_{a,b}$ by running the **backward mapping** on accuracies and correlations

---

The systems formed by the conditional independence relations differ based on whether $g$ maps to the Boolean hypercube $\{\pm 1\}^d$ or Euclidean space $\mathbb{R}^d$. In the latter case we obtain a system that has a simple closed form solution, detailed in Algorithm 2. In the discrete case (Algorithm 4, Appendix E), we need to use the quadratic formula to solve the system. We require an estimate of the prior $p$ on the label $y$; there are techniques do so (Anandkumar et al., 2014; Ratner et al., 2019); we tacitly assume we have access to it. The final step uses the backward mapping from mean to canonical parameters (Wainwright & Jordan, 2008). This approach is general, but it is easy in special cases: for Gaussians the canonical $\theta$ parameters are the inverse of the mean parameters.

**Performing Inference: Maximum Likelihood Estimator** Having estimated the label model canonical parameters $\hat{\theta}_a$ and $\hat{\theta}_{a,b}$ for all the sources, we use the maximum-likelihood estimator

$$\hat{y}_i = \underset{z \in \mathcal{Y}}{\arg\min} \; \frac{1}{m} \sum_{a=1}^m d_{\hat{\theta}_a}(\lambda^a(i), z) \tag{3}$$

Compare this approach to majority vote (2), observing that MV can produce arbitrarily bad outcomes. For example, suppose that we have $m$ LFs, one of which has a very high accuracy (e.g., large $\theta_1$) and the others very low accuracy (e.g., $\theta_2, \theta_3, \ldots, \theta_m = 0$). Then, $\lambda^1$ is nearly always correct, while the other LFs are nearly random. However, (2) weights them all equally, which will wash out the sole source of signal $\lambda^1$. On the other hand, (3) resolves the issue of equal weights on bad LFs by directly downweighting them.

**Simplifications** While the above models can handle very general scenarios, special cases are dramatically simpler. In particular, in the case of isotropic Gaussian errors, where $d_{\theta_a}(\lambda^a, y) = -\theta_a \|\lambda_a - y\|^2$, there is no need to perform an embedding, since we can directly rely on empirical averages like $\frac{1}{n}\sum_{i=1}^n d_{\theta_a}(\lambda^a(i), \lambda^b(i))$. The continuous triplet step simplifies to directly estimating the covariance entries $\hat{\theta}_a^{-1}$; the backward map is simply inverting this. More details can be found in Appendix D.

## 4 THEORETICAL ANALYSIS: ESTIMATION ERROR & CONSISTENCY

We show that, under certain conditions, Algorithm 1 produces consistent estimators of the mean parameters. We provide convergence rates for the estimators. As corollaries, we apply these to the settings of rankings and regression, where isometric embeddings are available. Finally, we give a bound on the inconsistency due to embedding distortion in the non-isometric case.

**Boolean Hypercube Case** We introduce the following finite-sample estimation error bound. It demonstrates consistency for mean parameter estimation for a triplet of LFs when using Algorithm 1. To keep our presentation simple, we assume: (i) the class balance $P(Y = y)$ are known, (ii) there are two possible values

of $Y$, $y_1$ and $y_2$, (iii) we can correctly recover signs (see Ratner et al. (2019)), (iv) we can find at least three conditionally independent labeling functions that can form a triplet, (v) the embeddings are isometric.

**Theorem 4.1.** *For any $\delta > 0$, for some $y_1$ and $y_2$ with known class probabilities $p = P(Y = y_1)$, the quadratic triplet method recovers $\alpha_i = P(g(\lambda^a)_i = 1|Y = y), \beta_i = P(g(\lambda^b)_i = 1|Y = y), \gamma_i = P(g(\lambda^c)_i = 1|Y = y)$ up to error $O((\frac{\ln(2d^2/\delta)}{2n})^{1/4})$ with probability at least $1 - \delta$ for any conditionally independent label functions $\lambda_a, \lambda_b, \lambda_c$ and for all $i \in [d]$.*

---

**Algorithm 2:** CONTINUOUSTRIPLETS

**Input:** Estimates $\hat{e}_{a,b}, \hat{e}_{a,c}, \hat{e}_{b,c}$, prior $p$, index $i$
Obtain variance $\mathbb{E}g(Y)_i^2$ from prior $p$
$\hat{\mathbb{E}}[g(\lambda^a)g(y)] \leftarrow \sqrt{|\hat{e}_{a,b}| \cdot |\hat{e}_{a,c}| \cdot \mathbb{E}[Y^2] / |\hat{e}_{b,c}|}$
$\hat{\mathbb{E}}[g(\lambda^b)g(y)] \leftarrow \sqrt{|\hat{e}_{a,b}| \cdot |\hat{e}_{b,c}| \cdot \mathbb{E}[Y^2] / |\hat{e}_{a,c}|}$
$\hat{\mathbb{E}}[g(\lambda^c)g(y)] \leftarrow \sqrt{|\hat{e}_{a,c}| \cdot |\hat{e}_{b,c}| \cdot \mathbb{E}[Y^2] / |\hat{e}_{a,b}|}$
**return** Accuracies
$\hat{\mathbb{E}}[g(\lambda^a)g(y)], \hat{\mathbb{E}}[g(\lambda^b)g(y)], \hat{\mathbb{E}}[g(\lambda^c)g(y)]$

---

We state our result via terms like $\alpha_i = P(g(\lambda^a)_i|Y = y)$; these can be used to obtain $\hat{\mathbb{E}}[g(\lambda^a)_i|y]$ and so recover $\hat{\mathbb{E}}[d_{\mathcal{Y}}(\lambda^a, y)]$.

To showcase the power of this result, we apply it to rankings from the symmetric group $S_\rho$ equipped with the Kendall tau distance $d_\tau$ (Kendall, 1938). This estimation problem is more challenging than learning the conventional Mallows model (Mallows, 1957)—and the standard Kemeny rule (Kemeny, 1959) used for rank aggregation will fail if applied to it. Our result yields consistent estimation when coupled with the aggregation step (3).

We use the isometric embedding $g$ into $\{\pm 1\}^{\binom{\rho}{2}}$: For $\pi \in S_\rho$, each entry in $g(\pi)$ corresponds to a pair $(i, j)$ with $i < j$, and this entry is 1 if in $\pi$ we have $i < j$ and $-1$ otherwise. We can show for $\pi, \gamma \in S_\rho$ that $\sum_{i=1} g(\pi)_i g(\gamma)_i = \binom{\rho}{2} - 2d_\tau(\pi, \gamma)$, and so recover $\hat{\mathbb{E}}[g(\lambda^a)_i g(y)_i]$, and thus $\hat{\mathbb{E}}[d_\tau(\lambda^a, y)]$. Then,

**Corollary 4.1.1.** *For any $\delta > 0$, $U > 0$, a prior over $y_1$ and $y_2$ with known class probability $p$, and using Algorithm 1 and Algorithm 4, for any conditionally independent triplet $\lambda^a, \lambda^b, \lambda^c$, with parameters $U > \theta_a, \theta_b, \theta_c > 4\ln(2)$, we can recover $\theta_a, \theta_b, \theta_c$ up to error $O(g_2^{-1}(\theta + (\log(2\rho^2)/(2\delta n))^{1/4}) - \theta)$ with probability at least $1 - \delta$, where $g_2(U) = (-\rho e^{-U})/((1 - e^{-U})^2) + \sum_{j=1}^{\rho} (j^2 e^{-Uj})/((1 - e^{-Uj})^2)$.*

Note that the error goes to zero as $O(n^{-1/4})$. This is an outcome of using the quadratic system, which is needed for generality. In the easier cases, where the quadratic approach is not needed (including the regression case), the rate is $O(n^{-1/2})$. Next, note that the scaling in terms of the embedding dimension $d$ is $O((\log(d))^{1/4})$—this is efficient. The $g_2$ function captures the cost of the backward mapping.

**Euclidean Space $\mathbb{R}^d$ Case** The following is an estimation error bound for the continuous triplets method in regression, where we use the squared Euclidean distance. The result is for $d = 1$ but can be easily extended.

**Theorem 4.2.** *Let $\hat{\mathbb{E}}[g(\lambda^a)g(y)]$ be an estimate of the accuracies $\mathbb{E}[g(\lambda^a)g(y)]$ using $n$ samples, where all LFs are conditionally independent given $Y$. If the signs of $a$ are recoverable, then with high probability*

$$\mathbb{E}[||\hat{\mathbb{E}}[g(\lambda^a)g(y)] - \mathbb{E}[g(\lambda^a)g(y)]||_2] = O\left(\left(a_{|min|}^{-10} + a_{|min|}^{-6}\right)\sqrt{\max(e_{max}^5, e_{max}^6)/n}\right).$$

*Here, $a_{|min|} = \min_i |\mathbb{E}[g(\lambda^i)g(y)]|$ and $e_{max} = \max_{j,k} e_{j,k}$.*

The error tends to zero with rate $O(n^{-1/2})$ as expected.

**Distortion & Inconsistency Bound** Next, we show how to control the inconsistency in parameter estimation as a function of the distortion. We write $\theta$ to be the vector of the canonical parameters, $\theta'$ for their distorted counterparts (obtained with a consistent estimator on the embedded distances), and $\mu, \mu'$ be the corresponding mean parameters. Let $1 - \varepsilon \leq d_g(g(y), g(y'))/d_{\mathcal{Y}}(y, y') \leq 1$ for all $y, y' \in \mathcal{Y} \times \mathcal{Y}$. Then, for a constant $e_{\min}$ (the value of which we characterize in the Appendix).

**Theorem 4.3.** *The inconsistency in estimating $\theta$ is bounded as $\|\theta - \theta'\| \leq \varepsilon\|\mu\|/e_{\min}$.*

## 5 EXPERIMENTS

We evaluated our universal approach with four sample applications, all new to WS: learning to rank, regression, learning in hyperbolic space, and estimation in generic metric spaces given by random graphs.

Our hypothesis is that the universal approach is capable of learning a label model and producing high-quality pseudolabels with the following properties:

- The quality of the pseudolabels is a function of the *number* and *quality* of the available sources, with more high-quality, independent sources yielding greater pseudolabel quality,
- Despite pseudolabels being noisy, an end model trained on *more* pseudolabels can perform as well or better than a fully supervised model trained on *fewer* true labels,
- The label model and inference procedure (3) improves on the majority vote equivalent—but only when LFs are of varying quality; for LFs of similar quality, the two will have similar performance.

Additionally, we expect to improve on naive applications of existing approaches that do not take structure into account (such as using Snorkel (Ratner et al., 2018) by mapping permutations to integer classes).

**Application I: Rankings** We applied our approach to obtain pseudorankings to train a downstream ranking model. We hypothesize that given enough signal, the produced pseudorankings can train a higher-quality model than using a smaller proportion of true labels. We expect our method produces better performance than the Snorkel baseline where permutations are converted into multi-class classification. We also anticipate that our inference procedure (3) improves on the MV baseline (2) when LFs have differing accuracies.

**Approach, Datasets, Labeling Functions, and End Model** We used the isotropic simplification of Algorithm 1. For inference, we applied (3). We compared against baseline fully-supervised models with only a proportion of the true labels (e.g., 20%, 50%, ...), the Snorkel framework (Ratner et al., 2018) converting rankings into classes, and majority vote (2). We used real-world datasets compatible with multiple label types, including a movies dataset and the BoardGameGeek dataset (2017) (BGG), along with synthetic data. For our movies dataset, we combined IMDb, TMDb, Rotten Tomatoes, and MovieLens movie review data to obtain features and weak labels. In Movies dataset, rankings were generated by picking $d = 5$ film items and producing a ranking based on their tMDb average rating. In BGG, we used the available rankings.

We created both real and simulated LFs. For simulated LFs, we sampled 1/3 of LFs from less noisy Mallows model, 2/3 of LFs from very noisy Mallows model. Details are in the Appendix H. For real LFs, we built labeling functions using external KBs as WS sources based on alternative movie ratings along with popularity, revenue, and vote count-based LFs. For the end model, we used PTRanking (Yu, 2020), with ListMLE (Xia et al., 2008) loss. We report the Kendall tau distance ($d_\tau$).

**Results** Figure 2 reports end model performance for the two datasets with varying numbers of simulated LFs. We observe that (i) as few as 12 LFs are sufficient to improve on a fully-supervised model trained on less data (as much as 20% of the dataset) and that (ii) as more LFs are added, and signal improves, performance also improves—as expected. Crucially, the Snorkel baseline, where rankings are mapped into classes, cannot perform as well as the universal approach; it is not meant to be effective general label settings. Table 2 shows the results when using real LFs, some good, some bad, constructed from alternative ratings and simple heuristics. Alternative ratings are quite accurate: MV and our method perform similarly. However, when poorer-quality LFs are added, MV rule tends to degrade more than our proposed model, as we anticipated.

**Application II: Regression** We used universal WS in the regression setting. We expect that with more LFs, we can obtain increasingly high-quality pseudolabels, eventually matching fully-supervised baselines.

**Approach, Datasets, Labeling Functions, End Model, Results** We used Algorithm 1, which uses the continuous triplets approach 2. For inference, we used the Gaussians simplification. As before, we compared

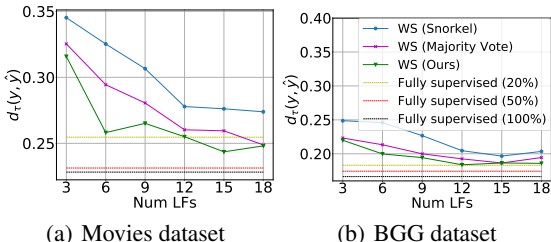

(a) Movies dataset     (b) BGG dataset         (a) Parameter Estimation Error

Figure 2: End model performance with ranking LFs (Left: Movies, Right: BGG). Training a model on pseudolabels is compared to fully-supervised baselines on varying proportions of the dataset along with the Snorkel baseline. Metric is the Kendall tau distance; lower is better.

Figure 3: Regression: parameter estimation error (left two plots) and label estimation error comparing to majority vote baseline (rightmost) with increasing number of samples.

| Setting | $d_\tau$ | MSE | Setting | $d_\tau$ | MSE |
|---|---|---|---|---|---|
| Fully supervised (10%) | 0.2731 | 0.3357 | WS (One LF, Rotten tomatoes) | 0.2495 | 0.4272 |
| Fully supervised (25%) | 0.2465 | 0.2705 | WS (One LF, IMDb score) | 0.2289 | 0.2990 |
| Fully supervised (50%) | 0.2313 | 0.2399 | WS (One LF, MovieLens score) | 0.2358 | 0.2690 |
| Fully supervised (100%) | 0.2282 | 0.2106 | | | |
| WS (3 LFs, MV (2)) | **0.2273** | 0.2754 | WS (3 scores + 3 bad LFs, MV (2)) | 0.2504 | - |
| WS (3 LFs, Ours) | **0.2274** | **0.2451** | WS (3 good + 3 bad LFs, Ours) | **0.2437** | - |

Table 2: End model performance with real-world rankings and regression LFs on movies. WS (3 scores, ·) shows the result of our algorithm combining 3 LFs. In ranking, high-quality LFs perform well (and better than fewer clean labels), but mixing in lower-quality LFs hurts majority vote (2) more than our proposed approach. In regression, our method yields performance similar to fully-supervised with 50% data, while outperforming MV.

against baseline fully-supervised models with a fraction of the true labels (e.g., 20%, 50%, ...) and MV (2). We used the Movies rating datasets with the label being the average rating of TMDb review scores across users and the BGG dataset with the label being the average rating of board games. We split data into 75% for training set, and 25% for the test set. For real-world LFs, we used other movie ratings in the movies dataset. Details are in the Appendix H for our synthetic LF generation procedure.For the end model, we used gradient boosting (Friedman, 2001). The performance metric is MSE.

Figure 9 (Appendix) shows the end model performance with WS compared to fully supervised on the movie reviews and board game reviews datasets. We also show parameter estimation error in Figure 3. As expected, our parameter estimation error goes down in the amount of available data. Similarly, our label estimator is consistent, while majority vote is not. Table 2 also reports the result of using real LFs for movies. Here, MV shows even worse performance than the best individual LF - Movie Lens Score. On the other side, our label model lower MSE than the best individual LF, giving similar performance with fully supervised learning with 50% training data.

**Application III: Geodesic Regression in Hyperbolic Space** Next, we evaluate our approach on the problem of geodesic regression on a Riemannian manifold $M$, specifically, a hyperbolic space with curvature $K = -50$. The goal of this task is analogous to Euclidean linear regression, except the dependent variable lies on a geodesic in hyperbolic space. Further background is in the Appendix H.

**Approach, Datasets, LFs, Results** We generate $y_i^*$ by taking points along a geodesic (a generalization of a line) parametrized by a tangent vector $\beta$ starting at $p$, further affecting them by noise. The objective of the end-model is to recover parameters $p$ and $\beta$, which is done using Riemannian gradient descent (RGD) to minimize the least-squares objective: $\hat{p}, \hat{\beta} = \arg\min_{q,\alpha} \sum_{i=1}^{n} d(\exp_q(x_i\alpha), y_i)^2$ where $d(\cdot, \cdot)$ is the hyperbolic distance. To generate each LF $\lambda_j$, we use noisier estimates of $y_i^*$, where the distribution is $\mathcal{N}(0, \sigma_j^2)$ and $\sigma_j^2 \sim \mathcal{U}_{[1.5+(15z_j), 4.5+(15z_j)]}$, $z_j \sim \text{Bernoulli}(0.5)$ to simulate heterogeneity across LFs;

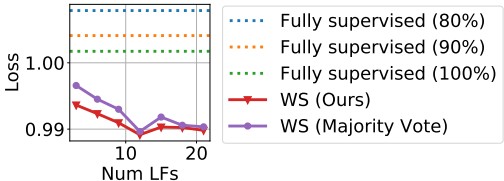

Figure 4: Comparison between our approach, (2), and fully-supervised in geodesic regression. Metric is least-squares objective; lower is better.

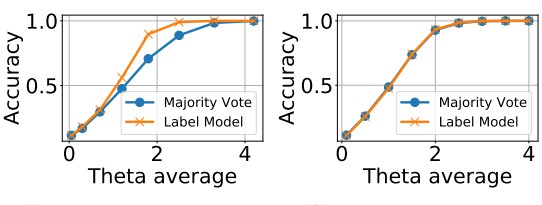

(a) Heterogeneous LFs    (b) Homogenous LFs

Figure 5: Comparison between our label model and majority voting in generic metric space. Metric is accuracy; higher is better.

the noise vectors are parallel transported to the appropriate space. For label model learning, we used the isotropic Gaussian simplification. Finally, inference used RGD to compute (3). We include MV (2) as a baseline. We compare fully supervised end models each with $n \in \{80, 90, 100\}$ labeled examples to weak supervision using only weak labels. In Figure 4 we see that the Fréchet mean baseline and our approach both outperform fully supervised geodesic regression despite the total lack of labeled examples. Intuitively, this is because with multiple noisier labels, we can produce a better pseudolabel than a single (but less noisy) label. As expected, our approach yields consistently lower test loss than the Fréchet mean baseline.

| Dataset/UAS | $\hat{Y}$ | $\lambda^1$ | $\lambda^2$ | $\lambda^3$ |
|---|---|---|---|---|
| cs_pdt-ud | **0.873** | 0.861 | 0.758 | 0.842 |
| en_ewt-ud | **0.795** | 0.792 | 0.733 | 0.792 |
| en_lines-ud | **0.850** | 0.833 | 0.847 | 0.825 |
| en_partut-ud | 0.866 | **0.869** | 0.866 | 0.817 |

Table 3: UAS scores for semantic dependency parsing. $\hat{Y}$ is synthesized from off-the-shelf parsing LFs.

**Application IV: Generic Metric Spaces** We also evaluated our approach on a structureless problem—a generic metric space generated by selecting random graphs $G$ with a fixed number of nodes and edges. The metric is the shortest-hop distance between a pair of nodes. We sampled nodes uniformly and obtain LF values with (1). Despite the lack of structure, we still anticipate that our approach will succeed in recovering the latent nodes $y$ when LFs have sufficiently high quality. We expect that the LM improves on MV (2) when LFs have heterogeneous quality, while the two will have similar performance on homogeneous quality LFs.

**Approach, Datasets, LFs, Results** We generated random graphs and computed the distance matrix yielding the metric. We used Algorithm (1) with isotropic Gaussian embedding and continuous triplets. For label model inference, we used (3). Figure 5 shows results on our generic metric space experiment. As expected, when LFs have a heterogeneous quality, LM yield better accuracy than MV. However, when labeling functions are of similar quality, LM and MV give similar accuracies.

**Application V: Semantic Dependency Parsing** We ran our technique on semantic dependency parsing tasks, using datasets in English and Czech from the Universal Dependencies collection (Nivre et al., 2020). The LFs are off-the-shelf parsers from Stanza (Qi et al., 2020) trained on different datasets in the same language. We model a space of trees with a metric given by the $\ell_2$ norm on the difference between the adjacency matrices. We measure the quality of the synthesized tree $\hat{Y}$ with the unlabeled attachment scores (UAS). Our results are shown in Table 3. As expected, when the parsers are of different quality, we can obtain a better result.

## 6  CONCLUSION

Weak supervision approaches allow users to overcome the manual labeling bottleneck for dataset construction. While successful, such methods do not apply to each potential label type. We proposed an approach to universally apply WS, demonstrating it for three applications new to WS: rankings, regression, and learning in hyperbolic space. We hope our proposed technique encourages applying WS to many more applications.

ACKNOWLEDGMENTS

We are grateful for the support of the NSF under CCF2106707 (Program Synthesis for Weak Supervision) and the Wisconsin Alumni Research Foundation (WARF).

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

The appendix contains additional details, proofs, and experimental results. The glossary contains a convenient reminder of our terminology (Appendix A). We provide a detailed related work section, explaining the context for our work (Appendix B). Next, we give the statement of the quadratic triplets algorithm (Appendix E). Afterwards, we give additional theoretical details: more details on the backward mapping, along with an extended discussion on the rankings problem. We continue with proofs of Theorem 4.1.1 and Theorem 4.2 (Appendix F). Finally, we give more details on the experimental setup (Appendix H) and additional experimental results including partial rankings (Appendix I, J).

## A  GLOSSARY

The glossary is given in Table 4 below.

| Symbol | Definition |
|---|---|
| $\mathcal{X}$ | feature space |
| $\mathcal{Y}$ | label metric space |
| $d_{\mathcal{Y}}$ | label metric |
| $x_1, x_2, \ldots, x_n$ | unlabeled datapoints from $\mathcal{X}$ |
| $y_1, y_2, \ldots, y_n$ | latent (unobserved) labels from $\mathcal{Y}$ |
| $s^1, s^2, \ldots, s^m$ | labeling functions / sources |
| $\lambda^1, \lambda^2, \ldots, \lambda^m$ | output of labeling functions |
| $n$ | number of data points |
| $m$ | number of LFs |
| $\lambda^a(i)$ | output of $a$th labeling function applied to $i$th sample $x_i$ |
| $\theta_a, \theta_{a,b}$ | canonical parameters in model (1) |
| $\mathbb{E}[d_a(\lambda^a, y)], \mathbb{E}[d_a(\lambda^a, \lambda^b)]$ | mean parameters in (1) |
| $g$ | injective mapping $g : \mathcal{Y} \to \mathbb{R}^d$ or $\{\pm 1\}^d$ |
| $\rho$ | number of items in ranking setting |
| $e_{a,b}$ | $E[g(\lambda^a)_i g(\lambda^b)_i]$ |
| $O_{a,b}$ | $P(g(\lambda^a)_i = 1, g(\lambda^b)_i = 1)$ |
| $l_a$ | $P(g(\lambda^a)_i = 1)$ |
| $S_\rho$ | symmetric group on $\{1, ..., \rho\}$ |
| $\pi$ | permutation $\in S_\rho$ |
| $d_\tau(\cdot, \cdot)$ | Kendall tau distance on permutations (Kendall, 1938) |

Table 4: Glossary of variables and symbols used in this paper.

## B  RELATED WORK

**Weak Supervision**   Existing weak supervision frameworks, starting with Ratner et al. (2016), select a particular model for the joint distribution among the sources and the latent true label, and then use the properties of the distribution to select an algorithm to learn the parameters. In Ratner et al. (2016), a factor model is used and a Gibbs-sampling based optimizer is the algorithm of choice. In Ratner et al. (2018), the model is a discrete Markov random field (MRF), and in particular, an Ising model. The algorithm used to learn the parameters solves a linear system obtained from a component of the inverse covariance matrix. In Sala et al. (2019) and Fu et al. (2020), the requirement to use the inverse covariance matrix is removed, and a set of systems among as few as three sources are used instead. These systems have closed-form solutions. All of these models could be represented within our framework. The quadratic triplets idea is described, but

not analyzed, in the appendix of Fu et al. (2020), applied to just the particular case of Ising models with singleton potentials. Our work uses these ideas and their extensions to the general setting, and provides theoretical analyses for the two sample applications we are interested in.

All of these papers refer to weak supervision frameworks; all of them permit the use of various types of information as labeling functions. For example, labeling functions might include crowdworkers Karger et al. (2011), distant supervision Mintz et al. (2009), co-training Blum & Mitchell (1998), and many others. Note that works like Dehghani et al. (2017) provide one type of weak supervision signal for rankings and can be used as a high-quality labeling function in our framework for the rankings case. That is, our work can integrate such sources, but does not directly compete with them.

The idea of learning notions like source accuracy and correlations, despite the presence of the latent label, is central to weak supervision. It has been shown up in other problems as well, such as crowdsourcing Dawid & Skene (1979); Karger et al. (2011) or topic modeling Anandkumar et al. (2014). Early approaches use expectation maximization (EM), but in the last decade, a number of exciting approaches have been introduced based on the method of moments. These include the tensor power method approach of Anandkumar et al. (2014) and its follow-on work, the explicitly crowdsourcing setting of Joglekar et al. (2013; 2015), and the graphical model learning procedures of Chaganty & Liang (2014); Raghunathan et al. (2016). Our approach can be thought of as an extension of some of these approaches to the more general setting we consider.

**Comparison With Existing Label Models**  There are several existing label models; these closely resemble (1) under particular specializations. For example, one of the models used for binary labels in Ratner et al. (2019); Varma et al. (2019) is the Ising model for $\lambda^1, \ldots, y \in \{\pm 1\}$

$$p(\lambda^1, \ldots, \lambda^m, y) = \frac{1}{Z} \exp \Big( \sum_{a=1}^{m} \theta_a \lambda^a y + \sum_{(a,b) \in E} \theta_{a,b} \lambda^a \lambda^b + \theta_Y y \Big). \tag{4}$$

A difference is that this is a joint model; it assumes $y$ is part of the model and adds a singleton potential prior term to it. Note that this model promotes agreement $\lambda^a y$ rather than penalizes disagreement $-\lambda^a d_{\mathcal{Y}}(\lambda^a, y)$, but this is nearly equivalent.

**Rankings and Permutations**  One of our chief applications is to rankings. There are several classes of distributions over permutations, including the Mallows model (Mallows, 1957) and the Plackett-Luce model (Plackett, 1975). We are particularly concerned with the Mallows model in this work, as it can be extended naturally to include labeling functions and the latent true label. Other generalizations include the generalized Mallows model (Marden, 2014) and the Mallows block model (Busa-Fekete et al., 2019). These generalizations, however, do not cover the heterogenous accuracy setting we are interested in.

A common goal is to learn the parameters of the model (the single parameter $\theta$ in the conventional Mallows model) and then to learn the central permutation that the samples are drawn from. A number of works in this direction include Caragiannis et al. (2016); Mukherjee (2016); Busa-Fekete et al. (2019). Our work extends results of this flavor to the generalization on the Mallows case that we consider. In order to learn the center permutation, estimators like the Kemeny rule (the procedure we generalize in this work) are used. Studies of the Kemeny rule include Kenyon-Mathieu & Schudy (2007); Caragiannis et al. (2016).

## C  EXTENSION TO MIXED LABEL TYPES

In the body of the paper, we discussed models for tasks where only one label type is considered. As a simple extension, we can operate with multiple label types. For example, this might include a classification task where weak label sources give their output as class labels and also might provide confidence values; that is, a pair of label types that include a discrete value and a continuous value. The main idea is to construct a finite

product space with individual label types. Suppose there are $k$ possible label types, i.e. $\mathcal{Y}_1, \mathcal{Y}_2, \cdots, \mathcal{Y}_k$. We construct the label space by the Cartesian product

$$\mathcal{Y} = \mathcal{Y}_1 \times \mathcal{Y}_2 \times \cdots \times \mathcal{Y}_k.$$

All that is left is to define the distance:

$$d_{\mathcal{Y}}^2(y_1, y_2) = \sum_{i=1}^{k} d_{\mathcal{Y}_i}^2(\mathrm{proj}_i(y_1), \mathrm{proj}_i(y_2)),$$

where $\mathrm{proj}_i$ is projection onto the $i$-th factor space. Then, using this combination, we extend the exponential family model (1), yielding

$$p(\lambda^1, \ldots, \lambda^m | y) = \frac{1}{Z} \exp \Big( \sum_{a=1}^{m} \sum_{i=1}^{k} -\theta_a^{(i)} d_{\mathcal{Y}_i}(\mathrm{proj}_i(\lambda^a), \mathrm{proj}_i(y))$$
$$+ \sum_{(a,b) \in E} \sum_{i=1}^{k} -\theta_{a,b}^{(i)} d_{\mathcal{Y}_i}(\mathrm{proj}_i(\lambda^a), \mathrm{proj}_i(\lambda^b)) \Big).$$

We can learn the parameters $\theta_a^{(i)}, \theta_{a,b}^{(i)}$ by Algorithm 1 using the same approach. Similarly, the inference procedure (3) can be extended to

$$\hat{y}_j = \arg\min_{z \in \mathcal{Y}} \sum_{a=1}^{m} \sum_{i=1}^{k} d_{\hat{\theta}_a^{(i)}}(\mathrm{proj}_i(\lambda^a(j)), \mathrm{proj}_i(z)).$$

There are two additional aspects worth mentioning. First, the user may wish to consider the scale of distances in each label space, since the scale of one of the factor spaces' distance might be dominant. To weight each label space properly, we can normalize each label space's distance. Second, we may wish to consider the abstention symbol as an additional element in each space. This permits LFs to output one or another type of label without necessarily emitting a full vector. This can also be handled; the underlying algorithm is simply modified to permit abstains as in Fu et al. (2020).

## D  UNIVERSAL LABEL MODEL FOR ISOTROPIC GAUSSIAN EMBEDDING

We illustrate the isotropic Gaussian version of the label model. While simple, it captures all of the challenges involved in label model learning, and it performs well in practice. The steps are shown in Algorithm 3.

Why does Algorithm 3 obtain the estimates of $\theta$ without observing the true label $y$? To see this, first, note that post-embedding, the model we are working with is given by

$$p(\lambda^1, \ldots, \lambda^m | y) = \frac{1}{Z} \exp \Big( \sum_{a=1}^{m} -\theta_a \|g(\lambda^a) - g(y)\|^2 + \sum_{(a,b) \in E} -\theta_{a,b} \|g(\lambda^a) - g(\lambda^b)\|^2 \Big).$$

If the embedding is a bijection, the resulting model is indeed is a multivariate Gaussian. Note that the term "isotropic" here refers to the fact that for $d > 1$, the covariance term for the random vector $\lambda^a$ is a multiple of $I$.

Now, observe that if $(a, b) \notin E$, we have that $\mathbb{E}\left[ \|\lambda^a - \lambda^b\|^2 \right] = \mathbb{E}\left[ \|(\lambda^a - y) - (\lambda^b - y)\|^2 \right] = \mathbb{E}\left[ \|\lambda^a - y\|^2 \right] + \mathbb{E}\left[ \|\lambda^b - y\|^2 \right]$. Note that we can estimate the left-hand side $\mathbb{E}\left[ \|\lambda^a - \lambda^b\|^2 \right]$ from samples,

---

**Algorithm 3:** Isotropic Gaussian Label Model Learning

---

1: **Input:** Output of labeling functions $\lambda^a(i)$, correlation set $E$
2: **for** $a \in \{1, 2, \ldots, m\}$ **do**
3:     **for** $b \in \{1, 2, \ldots, m\} \setminus a$ **do**
4:         **Estimate Correlations:**$\forall i, j, \hat{\mathbb{E}}\left[d(\lambda^a, \lambda^b)\right] \leftarrow \frac{1}{n}\sum_{t=1}^{n} d(\lambda^a(i), \lambda^b(i))$
5:     **end for**
6:     **Estimate Accuracy:** Pick $b, c : (a, b) \notin E, (a, c) \notin E, (b, c) \notin E$
        $\hat{\mathbb{E}}\left[d(\lambda^a, y)\right] \leftarrow 1/2(\mathbb{E}\left[d(\lambda^a, \lambda^b)\right] + \mathbb{E}\left[d(\lambda^a, \lambda^c)\right] - \mathbb{E}\left[d(\lambda^b, \lambda^c)\right])$
7:     Form estimated covariance matrix $\hat{\Sigma}$ from accuracies and correlations; Compute $\hat{\theta} \leftarrow \hat{\Sigma}^{-1}$
8: **end for**
9: **return** $\hat{\theta}_a, \hat{\theta}_{a,b}$

---

while the right-hand side contains two of our accuracy terms. We can then form two more equations of this type (involving the pairs $a, c$ and $b, c$) and solve the resulting linear system. Concretely, we have that

$$\mathbb{E}\left[\|g(\lambda^a) - g(\lambda^b)\|^2\right] = \mathbb{E}\left[\|g(\lambda^a) - y\|^2\right] + \mathbb{E}\left[\|g(\lambda^b) - y\|^2\right]$$
$$\mathbb{E}\left[\|g(\lambda^a) - g(\lambda^c)\|^2\right] = \mathbb{E}\left[\|g(\lambda^a) - y\|^2\right] + \mathbb{E}\left[\|g(\lambda^c) - y\|^2\right]$$
$$\mathbb{E}\left[\|g(\lambda^b) - g(\lambda^c)\|^2\right] = \mathbb{E}\left[\|g(\lambda^b) - y\|^2\right] + \mathbb{E}\left[\|g(\lambda^c) - y\|^2\right].$$

To obtain the estimate of $\mathbb{E}\left[\|g(\lambda^a) - y\|^2\right]$, we add the first two equations, subtract the third, and divide by two.This produces the accuracy estimation step in Algorithm 3. To obtain the estimate of the canonical parameters $\theta$, it is sufficient to invert our estimate of the covariance matrix. In practice, note that we need not actually compute an embedding; we can directly work with the original metric space distances by implicitly assuming that there exists an isometric embedding.

# E    ADDITIONAL ALGORITHMIC DETAILS

**Quadratic Triplets**    We give the full statement of Algorithm 4, which is the accuracy estimation algorithms for the case $Im(g) = \{\pm1\}^d$.

**Inference simplification in** $\mathbb{R}$    For the simplification of inference in the isotropic Gaussian embedding, we do not even need to recover the canonical parameters; it is enough to use the mean parameters estimated in the label model learning step. For example, if $d = 1$, we can write these accuracy mean parameters as $\hat{\mathbb{E}}\left[g(\lambda^i)g(y)\right]$ and put them into a vector $\hat{\mathbb{E}}\left[\Sigma\right]_{\Lambda y}$. The correlations can be placed into the sample covariance matrix $\hat{\Sigma}_{g(\lambda^1),\ldots,g((\lambda^m)}$. Then,

$$\hat{y}(i) := \mathbb{E}\left[y|\lambda^1(i), \ldots \lambda^m(i)\right] = \hat{\Sigma}_{\Lambda y}^T \hat{\Sigma}_{\Lambda}^{-1}[\lambda^1(i), \ldots, \lambda^m(i)]. \tag{5}$$

This is simply the mean of a conditional (scalar) Gaussian.

# F    ADDITIONAL THEORY DETAILS

We discuss further details for several theoretical notions. The first provides more details on how to obtain the canonical accuracy parameters $\theta_a$ from the mean parameters $\mathbb{E}\left[d_\tau(\lambda^a, y)\right]$ in the rankings case. The second involves a discussion of learning to rank problems, with additional details on the weighted estimator (3).

---

**Algorithm 4:** QUADRATICTRIPLETS

---

**Input:** Estimates $O_{a,b}, O_{a,c}, O_{b,c}, \ell_a, \ell_b, \ell_c$, prior $p$, index $i$

**for** $y$ in $\mathcal{Y}$ **do**

    Obtain probability $p' = P(Y = y)$ from prior $p$

    Set $\beta \leftarrow (O_{a,b}(1 - p') + (\ell_a - pz)\ell_b)/(p'z - p'\ell_a)$

    Set $\gamma \leftarrow (O_{a,c}(1 - p') + (\ell_a - pz)\ell_c)/(p'z - p'\ell_a)$

    Solve quadratic $(p\beta\gamma + \ell^b\ell^c - p\beta\ell^c - p\gamma\ell^b - O_{a,b}(1-p))(p'\alpha - p'\ell_a)^2 = 0$ in $z$

    $\hat{P}(g(\lambda^a)_i|Y = y) \leftarrow z$

    $\hat{P}(g(\lambda^b)_i|Y = y) \leftarrow (O_{a,b}(1 - p') + (\ell_a - p\hat{P}(g(\lambda^a)_i|Y = y))\ell_b)/(p'\hat{P}(g(\lambda^a)_i|Y = y) - p'\ell_a)$

    $\hat{P}(g(\lambda^c)_i|Y = y) \leftarrow (O_{a,c}(1 - p') + (\ell_a - p\hat{P}(g(\lambda^a)_i|Y = y))\ell_c)/(p'\hat{P}(g(\lambda^a)_i|Y = y) - p'\ell_a)$

**end for**

**return** Accuracies $\hat{P}(g(\lambda^a)_i|Y = y), \hat{P}(g(\lambda^b)_i|Y = y), \hat{P}(g(\lambda^c)_i|Y = y)$

---

**More on the backward mapping for rankings** We note, first, that the backward mapping is quite simple for the Gaussian cases in $\mathbb{R}^d$: it just involves inverting the mean parameters. As an intuitive example, note that the canonical parameter in the interior of the multivariate Gaussian density is $\Sigma^{-1}$, the inverse covariance matrix. The more challenging aspect is to deal with the discrete setting, and, in particular, its application to rankings. We do so below.

We describe how we can recover the canonical accuracy parameter $\theta_a$ for a label function $\lambda^a$ given accuracy estimates $P(g(\lambda^a)_i = 1|Y = y)$ for all $y \in \mathcal{Y}$. By equation (1), the marginal distribution of $\lambda^a$ is specified by

$$p(\lambda^a) = \frac{1}{Z}\exp(-g(\lambda^a)^T\theta_a g(y)). \tag{6}$$

Since this is an exponential model, it follows from Fligner & Verducci (1986) that

$$\mathbb{E}[D] = \frac{d\log(M(t))}{dt}\bigg|_{t=-\theta_a}, \tag{7}$$

where $D = \sum_i \mathbb{1}\{g(\lambda^a)_i \neq g(y)_i\}$ and $M(t)$ is the moment generating function of $D$ under (5). $\mathbb{E}[D]$ can be easily estimated from the accuracy parameters obtained from the triplet algorithm, and the inverse of (6) can then be solved for. For instance, in the rankings case, it can be shown (Fligner & Verducci, 1986) that

$$M(-\theta) = \frac{1}{\rho!}Z(\theta).$$

Additionally, we have that the partition function satisfies (Chierichetti et al., 2014)

$$Z(\theta) = \prod_{j \leq k} \frac{1 - e^{\theta j}}{1 - e^{\theta}}.$$

It follows that

$$M(t) = \frac{1}{k!}\prod_{j \leq k} \frac{1 - e^{\theta j}}{1 - e^{\theta}}. \tag{8}$$

Using this, we can then solve for (6) numerically.

**Rank aggregation and the weighted Kemeny estimator** Next, we provide some additional details on the learning to rank problem. The model (1) without correlations can be written

$$p(\lambda^1, \ldots, \lambda^m|y) = \frac{1}{Z}\exp\left(\sum_a -\theta_a d_\tau(\lambda^a, y)\right). \tag{9}$$

Thus, if we only have one labeling function, we obtain the Mallows model (Mallows, 1957) for permutations, whose standard form is $p(\lambda^1|y) = 1/Z \exp(-\theta_1 d_\tau(\lambda^1, y))$. Permutation learning problems often use the Mallows model and its relatives. The permutation $y$ (called the *center*) is fixed and $n$ samples of $\lambda^1$ are drawn from the model. The goal is to (1) estimate the parameter $\theta_1$ and (2) estimate the center $y$. This neatly mirrors our setting, where (1) is label model learning and (2) is the inference procedure.

However, our problem is harder in two ways. While we do have $n$ samples from each marginal model with parameter $\theta_i$, these are for different centers $y_1, \ldots, y_n$—so we cannot aggregate them or directly estimate $\theta_i$. That is, for LF $a$, we get to observe $\lambda^a(1), \lambda^a(2), \ldots, \lambda^a(n)$. However, these all come from different conditional models $P(\cdot|y_i)$, with a different $y_i$ for each draw. In the uninteresting case where the $y_i$'s are identical, we obtain the standard setting. On the other hand, we do get to observe $m$ views (from the LFs) of the same permutation $y_i$. But, unlike in standard rank aggregation, these do not come from the same model—the $\theta_a$ accuracy parameters differ in (9). However, if $\theta_a$ is identical (same accuracy) for all LFs, we get back to the standard case. Thus we can recover the standard permutation learning problem in the special cases of identical labels or identical accuracies—but such assumptions are unlikely to hold in practice.

Note that our inference procedure (3) is the weighted version of the standard Kemeny procedure used for rank aggregation. Observe that this is the maximum likelihood estimator on the model (1), specialized to permutations. This is nearly immediate: maximizing the linear combination (where the parameters are weights) produces the smallest negative term on the inside of the exponent above.

Next, we note that it is possible to show that the sample complexity of learning the permutation $y$ using this estimator is still $\Theta(\log(m/\varepsilon))$ by using the pairwise majority result in Caragiannis et al. (2016).

Finally, a brief comment on computational complexity: it is known that finding the minimizer of the Kemeny rule (or our weighted variant) is NP-hard (Davenport & Kalagnanam, 2004). However, there are PTAS available for it (Kenyon-Mathieu & Schudy, 2007). In practice, our permutations are likely to be reasonably short (for example, in our experiments, we typically use length 5) so that we can directly perform the minimization. In cases with longer permutations, we can rely on the PTAS.

## G    PROOF OF THEOREMS

Next we give the proofs of the proposition and our two theorems, restated for convenience.

### G.1    DISTORTION BOUND

Our result that captures the impact of distortion on parameter error is

**Theorem 4.3.** *The inconsistency in estimating $\theta$ is bounded as $\|\theta - \theta'\| \leq \varepsilon \|\mu\|/e_{\min}$.*

Before we begin the proof, we restate, in greater detail, some of our notation. We write $p_{\theta;d_\mathcal{Y}}$ for the true model

$$p_{\theta;d_\mathcal{Y}}(\lambda^1, \ldots, \lambda^m|y) = \frac{1}{Z} \exp \Big( \sum_{a=1}^m -\theta_a d_\mathcal{Y}(\lambda^a, y) + \sum_{(a,b)\in E} -\theta_{a,b} d_\mathcal{Y}(\lambda^a, \lambda^b) \Big)$$

and $p_{\theta';d_g}$ for models that rely on distances in the embedding space

$$p_{\theta;d_g}(\lambda^1, \ldots, \lambda^m|y) = \frac{1}{Z} \exp \Big( \sum_{a=1}^m -\theta'_a d_g(g(\lambda^a), g(y)) + \sum_{(a,b)\in E} -\theta'_{a,b} d_g(g(\lambda^a), g(\lambda^b)) \Big).$$

We also write $\theta$ to be the vector of the canonical parameters $\theta_a$ and $\theta_{a,b}$ in $p_{\theta;d_\mathcal{Y}}$ and $\theta'$ to be its counterpart for $p_{\theta';d_g}$. Let $\mu$ be the vector of mean parameters whose terms are $\mathbb{E}[d_\mathcal{Y}(\lambda^a, y)]$ and $\mathbb{E}[d_\mathcal{Y}(\lambda^a, \lambda^b)]$.

Similarly, we write $\mu'$ for the version given by $p_{\theta';d_g}$. Let $\Theta$ be a subspace so that $\theta, \theta' \in \Theta$. Since our exponential family is minimal, we have that the log partition function $A(\tilde{\theta})$ has the property that $\nabla^2 A(\tilde{\theta})$ is the covariance matrix and is positive definite. Suppose that the smallest eigenvalue of $\nabla^2 A(\tilde{\theta})$ for $\tilde{\theta} \in \Theta$ is $e_{\min}$.

For our embedding, suppose we first normalize (ie, divide by a constant) our embedding function so that

$$\frac{d_g(g(y), g(y'))}{d_{\mathcal{Y}}(y, y')} \le 1$$

for all $y, y' \in \mathcal{Y} \times \mathcal{Y}$. In other words, our embedding function $g$ never expands the distances in the original metric space, but, of course, it may compress them. The distortion measures how bad the compression can be, Let $\varepsilon$ be the smallest value so that

$$1 - \varepsilon \le \frac{d_g(g(y), g(y'))}{d_{\mathcal{Y}}(y, y')} \le 1$$

for all $y, y' \in \mathcal{Y} \times \mathcal{Y}$. If $g$ is isometric, then we obtain $\varepsilon = 0$. On the other hand, as $\varepsilon$ approaches 1, the amount of compression can be arbitrarily bad.

*Proof.* We use Lemma 8 from Fu et al. (2020). It states that

$$\|\theta - \theta'\| \le \frac{1}{e_{\min}} \|\mu - \mu'\|$$
$$\le \frac{\varepsilon}{e_{\min}} \|\mu\|.$$

To see the latter, we note that $\|\mu - \mu'\| \le \|\mu(1 - \frac{\mu'}{\mu})\| \le \|\mu\|(1 - (1 - \varepsilon)) = \|\mu\|\varepsilon$, where $\frac{\mu'}{\mu}$ is the element-wise ratios, and where we applied our distortion bound. $\square$

### G.2 BINARY HYPERCUBE CASE

**Theorem 4.1.** *For any $\delta > 0$, for some $y_1$ and $y_2$ with known class probabilities $p = P(Y = y_1)$, the quadratic triplet method recovers $\alpha_i = P(g(\lambda^a)_i = 1 | Y = y), \beta_i = P(g(\lambda^b)_i = 1 | Y = y), \gamma_i = P(g(\lambda^c)_i = 1 | Y = y)$ up to error $O((\frac{\ln(2d^2/\delta)}{2n})^{1/4})$ with probability at least $1 - \delta$ for any conditionally independent label functions $\lambda_a, \lambda_b, \lambda_c$ and for all $i \in [d]$.*

*Proof.* Define

$$\mu_i^a = \begin{bmatrix} P(g(\lambda^a)_i = 1 | Y = y) & P(g(\lambda^a)_i = 1 | Y \ne y) \\ P(g(\lambda^a)_i = -1 | Y = y) & P(g(\lambda^a)_i = -1 | Y \ne y) \end{bmatrix}, \quad P = \begin{bmatrix} P(Y = y) & 0 \\ 0 & P(Y \ne y) \end{bmatrix},$$

and

$$O_i^{ab} = \begin{bmatrix} P(g(\lambda^a)_i = 1, g(\lambda^b)_i = 1) & P(g(\lambda^a)_i = 1, g(\lambda^b)_i = -1) \\ P(g(\lambda^a)_i = -1, g(\lambda^b)_i = 1) & P(g(\lambda^a)_i = -1, g(\lambda^b)_i = -1) \end{bmatrix}.$$

By conditional independence, we have that

$$\mu_i^a P (\mu_i^b)^T = O_i^{ab}. \tag{10}$$

Note that we can express

$$P(g(\lambda^a)_i = 1 | Y \ne y) = \frac{P(g(\lambda^a)_i = 1)}{P(Y \ne y)} - \frac{P(g(\lambda^a)_i = 1 | Y = y) P(Y = y)}{P(Y \ne y)}.$$

We can therefore rewrite the top row of $\mu_i^a$ as $[\alpha_i, q_i^a - r\alpha_i]$ where $q_i^a = \frac{P(g(\lambda^a)_i = 1)}{P(Y \neq y)}$ and $r = \frac{P(Y=y)}{P(Y \neq y)}$. After estimating the entries of $O_i$'s and $q_i$'s, we consider the top-left entry of (10) for every pair of $a, b$ and $c$, to get the following system of equations

$$t\alpha_i\beta_i + \hat{q}_i^a \hat{q}_i^b - \hat{q}_i^a r\beta_i - \hat{q}_i^b r\alpha_i = \frac{\hat{O}_i^{ab}}{1-p},$$

$$t\alpha_i\gamma_i + \hat{q}_i^a \hat{q}_i^c - \hat{q}_i^a r\gamma_i - \hat{q}_i^c r\alpha_i = \frac{\hat{O}_i^{ac}}{1-p},$$

$$t\beta_i\gamma_i + \hat{q}_i^b \hat{q}_i^c - \hat{q}_i^b r\gamma_i - \hat{q}_i^c r\beta_i = \frac{\hat{O}_i^{bc}}{1-p},$$

where $\beta_i$ and $\gamma_i$ are the top-left entries of $\mu_i^b$ and $\mu_i^c$, respectively, $p = P(Y = y)$, and $t = \frac{p}{(1-p)^2}$.

For ease of notation, we write $\hat{q}_i^a$ as $\hat{q}_a$ and so on. Rearranging the first and third equations gives us expressions for $\alpha$ and $\gamma$ in terms of $\beta$.

$$\alpha = \frac{\frac{\hat{O}_{ab}}{1-p} + \hat{q}_a r\beta - \hat{q}_a \hat{q}_b}{t\beta - \hat{q}_b r},$$

$$\gamma = \frac{\frac{\hat{O}_{bc}}{1-p} + \hat{q}_c r\beta - \hat{q}_b \hat{q}_c}{t\beta - \hat{q}_b r}.$$

Substituting these expressions into the second equation of the system gives

$$t\frac{\hat{q}_a r\beta + \frac{\hat{O}_{ab}}{1-p} - \hat{q}_a \hat{q}_b}{t\beta - \hat{q}_b r} \cdot \frac{\hat{q}_c r\beta + \frac{\hat{O}_{bc}}{1-p} - \hat{q}_b \hat{q}_c}{t\beta - \hat{q}_b r} + \hat{q}_a \hat{q}_c - \hat{q}_a r\frac{\hat{q}_c r\beta + \frac{\hat{O}_{bc}}{1-p} - \hat{q}_b \hat{q}_c}{t\beta - \hat{q}_b r} - \hat{q}_c r\frac{\hat{q}_a r\beta + \frac{\hat{O}_{ab}}{1-p} - \hat{q}_a \hat{q}_b}{t\beta - \hat{q}_b r} = \frac{\hat{O}_{ac}}{1-p}.$$

We can multiply the equation by $(t\beta - \hat{q}_b r)^2$ to get

$$t(\hat{q}_a r\beta + \frac{\hat{O}_{ab}}{1-p} - \hat{q}_a \hat{q}_b) \cdot (\hat{q}_c r\beta + \frac{\hat{O}_{bc}}{1-p} - \hat{q}_b \hat{q}_c) + \hat{q}_a \hat{q}_c((t\beta - \hat{q}_b r)^2)$$

$$-\hat{q}_a r(\hat{q}_c r\beta + \frac{\hat{O}_{bc}}{1-p} - \hat{q}_b \hat{q}_c) \cdot (t\beta - \hat{q}_b r) - \hat{q}_c r(\hat{q}_a r\beta + \frac{\hat{O}_{ab}}{1-p} - \hat{q}_a \hat{q}_b) \cdot (t\beta - \hat{q}_b r)$$

$$= \frac{\hat{O}_{ac}}{1-p} \cdot (t\beta - \hat{q}_b r)^2$$

$$\Rightarrow t\left(\hat{q}_a \hat{q}_c r^2 \beta^2 + (\frac{\hat{O}_{ab}}{1-p} - \hat{q}_a \hat{q}_b)\hat{q}_c r\beta + (\frac{\hat{O}_{bc}}{1-p} - \hat{q}_b \hat{q}_c)\hat{q}_a r\beta + (\frac{\hat{O}_{ab}}{1-p} - \hat{q}_a \hat{q}_b)(\frac{\hat{O}_{bc}}{1-p} - \hat{q}_b \hat{q}_c)\right)$$

$$+\hat{q}_a \hat{q}_c \left(t^2 \beta^2 - 2\hat{q}_b r t\beta + \hat{q}_b^2 r^2\right)$$

$$-\hat{q}_a r\left(\hat{q}_c r t\beta^2 + (\frac{\hat{O}_{bc}}{1-p} - \hat{q}_b \hat{q}_c)t\beta - \hat{q}_b \hat{q}_c r^2 \beta - \hat{q}_b r(\frac{\hat{O}_{bc}}{1-p} - \hat{q}_b \hat{q}_c)\right)$$

$$-\hat{q}_c r\left(\hat{q}_a r t\beta^2 + (\frac{\hat{O}_{ab}}{1-p} - \hat{q}_a \hat{q}_b)t\beta - \hat{q}_a \hat{q}_b r^2 \beta - \hat{q}_b r(\frac{\hat{O}_{ab}}{1-p} - \hat{q}_a \hat{q}_b)\right)$$

$$= \frac{\hat{O}_{ac}}{1-p} \cdot \left(t^2 \beta^2 - 2\hat{q}_b r t\beta + \hat{q}_b^2 r^2\right)$$

$$\Rightarrow \beta^2 \left[\hat{q}_a\hat{q}_c t^2 - \hat{q}_a\hat{q}_c r^2 e - \frac{\hat{O}_{ac}}{1-p}\cdot t^2\right]$$

$$+\beta\left[t\left((\frac{\hat{O}_{ab}}{1-p} - \hat{q}_a\hat{q}_b)\hat{q}_c r + (\frac{\hat{O}_{bc}}{1-p} - \hat{q}_b\hat{q}_c)\hat{q}_a r\right) - 2\hat{q}_a\hat{q}_c\hat{q}_b rt\right.$$

$$\left.-\hat{q}_a r\left((\frac{\hat{O}_{bc}}{1-p} - \hat{q}_b\hat{q}_c)t - \hat{q}_b\hat{q}_c r^2\right) - \hat{q}_c r\left((\frac{\hat{O}_{ab}}{1-p} - \hat{q}_a\hat{q}_b)t - \hat{q}_a\hat{q}_b r^2\right) + \frac{\hat{O}_{ac}}{1-p}\cdot\left(2\hat{q}_b rt\right)\right]$$

$$+\left[t(\frac{\hat{O}_{ab}}{1-p} - \hat{q}_a\hat{q}_b)(\frac{\hat{O}_{bc}}{1-p} - \hat{q}_b\hat{q}_c) + \hat{q}_a\hat{q}_c\hat{q}_b^2 r^2\right.$$

$$\left.+\hat{q}_a\hat{q}_b r^2(\frac{\hat{O}_{bc}}{1-p} - \hat{q}_b\hat{q}_c) + \hat{q}_b\hat{q}_c r^2(\frac{\hat{O}_{ab}}{1-p} - \hat{q}_a\hat{q}_b) - \frac{\hat{O}_{ac}}{1-p}\cdot\hat{q}_b^2 t^2\right] = 0.$$

The only sources of error are from estimating $c$'s and $O$'s. Let $\varepsilon_{c'}$ denote the error for the $c$'s. Let $\varepsilon_O$ denote the error from $O$'s. Applying the quadratic formula we obtain $\beta = (-b' \pm \sqrt{(b')^2 - 4a'c'})/(2a')$ where the coefficients are

$$a' = \hat{q}_a\hat{q}_c t^2 - \hat{q}_a\hat{q}_c r^2 t - \frac{\hat{O}_{ac}}{1-p}\cdot t^2$$

$$b' = t\left((\frac{\hat{O}_{ab}}{1-p} - \hat{q}_a\hat{q}_b)\hat{q}_c r + (\frac{\hat{O}_{bc}}{1-p} - \hat{q}_b\hat{q}_c)\hat{q}_a r\right) - 2\hat{q}_a\hat{q}_c\hat{q}_b rt$$

$$-\hat{q}_a r\left((\frac{\hat{O}_{bc}}{1-p} - \hat{q}_b\hat{q}_c)t - \hat{q}_b\hat{q}_c r^2\right) - \hat{q}_c r\left((\frac{\hat{O}_{ab}}{1-p} - \hat{q}_a\hat{q}_b)t - \hat{q}_a\hat{q}_b r^2\right)$$

$$+\frac{\hat{O}_{ac}}{1-p}\cdot\left(2\hat{q}_b rt\right)$$

$$c' = t(\frac{\hat{O}_{ab}}{1-p} - \hat{q}_a\hat{q}_b)(\frac{\hat{O}_{bc}}{1-p} - \hat{q}_b\hat{q}_c) + \hat{q}_a\hat{q}_c\hat{q}_b^2 r^2 + \hat{q}_a\hat{q}_b r^2(\frac{\hat{O}_{bc}}{1-p} - \hat{q}_b\hat{q}_c)$$

$$+\hat{q}_b\hat{q}_c r^2(\frac{\hat{O}_{ab}}{1-p} - \hat{q}_a\hat{q}_b) - \frac{\hat{O}_{ac}}{1-p}\cdot\hat{q}_b^2 r^2.$$

Let $\varepsilon_{a'}, \varepsilon_{b'}, \varepsilon_{c'}$ denote the error for each coefficient. That is, $\varepsilon_{a'} = |(a')^* - a'|$, where $(a')^*$ is the population-level coefficient, and similarly for $b', c'$. Let $\varepsilon_c$ and $\varepsilon_O$ indicate the estimation error for the $c$ and $O$ terms. Then

$$\varepsilon_{a'} = O\left(t^2 p\varepsilon_c + \frac{t^2}{1-p}\varepsilon_O\right),$$

$$\varepsilon_{b'} = O\left(\frac{rt}{1-p}\left(\varepsilon_c + \varepsilon_O\right)\right),$$

$$\varepsilon_{c'} = O\left(\left(\frac{t}{(1-p)^2} + \frac{r^2}{1-p} + r^2\right)\varepsilon_c + \left(\frac{t^2}{(1-p)^2} + \frac{r^2}{1-p}\right)\varepsilon_O\right).$$

Because $d$ and $e$ are functions of $p$, if we ignore the dependence on $p$, we get that

$$\varepsilon_{a'} = \varepsilon_{b'} = \varepsilon_{c'} = O(\varepsilon_c + \varepsilon_O).$$

Furthermore,

$$\varepsilon_{(b')^2} = O(\varepsilon_b), \varepsilon_{a'c'} = O(\varepsilon_a + \varepsilon_c).$$

It follows that

$$\varepsilon_\beta = O(\sqrt{\varepsilon_{c'} + \varepsilon_O}).$$

Next, note that $O$ and $c$ are both the averages of indicator variables, where the $c_i$'s involve $P(g(\lambda^a)_i = 1)$ and the $O_i^{ab}$'s upper-left corners compute $P(g(\lambda^a)_i = 1, g(\lambda^b)_i = 1)$. Thu we can apply Hoeffding's inequality and combine this with the union bound to bound the above terms. We have with probability at least $1 - \frac{\delta}{2}$, $\varepsilon_{c_i} \leq \sqrt{\frac{\log(2d/\delta)}{2n}}$ and similarly with probability at least $1 - \frac{\delta}{2}$, $\varepsilon_{O_i} \leq \sqrt{\frac{\log(2d/\delta)}{2n}}$ for all $i \in [d]$. It follows that with probability at least $1 - \delta$, $\varepsilon_\alpha = \varepsilon_\beta = \varepsilon_\gamma = O((\frac{\log(2d/\delta)}{2n})^{1/4})$. $\qquad\square$

We can now prove the main theorem in the rankings case.

**Corollary 4.1.1.** *For any $\delta > 0$, $U > 0$, a prior over $y_1$ and $y_2$ with known class probability $p$, and using Algorithm 1 and Algorithm 4, for any conditionally independent triplet $\lambda^a, \lambda^b, \lambda^c$, with parameters $U > \theta_a, \theta_b, \theta_c > 4\ln(2)$, we can recover $\theta_a, \theta_b, \theta_c$ up to error $O(g_2^{-1}(\theta + (\log(2\rho^2)/(2\delta n))^{1/4}) - \theta)$ with probability at least $1 - \delta$, where $g_2(U) = (-\rho e^{-U})/((1 - e^{-U})^2) + \sum_{j=1}^{\rho}(j^2 e^{-Uj})/((1 - e^{-Uj})^2)$.*

*Proof.* Consider a pair of items $(a, b)$. Without loss of generality, suppose $a \prec_{y_1} b$ and $a \succ_{y_2} b$. Define $\alpha_{a,b} = P(a \prec_{\lambda^a} b | y_1)$, $\alpha'_{a,b} = P(a \succ_{\lambda^a} b | y_2)$ then our estimate for $P(\lambda^a_{(a,b)} \sim Y_{(a,b)})$ where $\lambda^a_{(a,b)} \sim Y_{(a,b)}$ denotes the event that label function $i$ ranks $(a, b)$ correctly would be

$$\hat{P}(\lambda^a_{(a,b)} \sim Y_{(a,b)}) = p\hat{\alpha}_{a,b} + (1 - p)(1 - \hat{\alpha}'_{a,b})$$

which has error $O(\epsilon_\alpha)$. Then, note that $\mathbb{E}[d(\lambda^a, Y)] = \sum_{a,b} P(\lambda^a_{(a,b)} \sim Y_{(a,b)})$. Therefore, we can compute the estimate $\hat{\mathbb{E}}[d(\lambda^a, Y)] = \sum_{a,b} \hat{\mathbb{E}}[\lambda_i \sim Y]_{a,b}$ which has error $O(\binom{\rho}{2}(\frac{\ln(6)/\delta}{2n})^{1/4})$.

Recall from (7) we have that

$$\mathbb{E}_\theta[D] = \frac{d[\log(M(t))]}{dt}\bigg|_{t=-\theta},$$

where $M(t)$ is the moment generating function, and recall from (8) that

$$M(t) = \frac{1}{k!}\prod_{j \leq k}\frac{1 - e^{\theta j}}{1 - e^\theta}.$$

It follows that

$$\mathbb{E}_\theta[D] = \frac{ke^{-\theta}}{1 - e^{-\theta}} - \sum_{j \leq k}\frac{je^{-\theta j}}{1 - e^{-\theta j}} \tag{11}$$

$$\Rightarrow \frac{d}{d\theta}\mathbb{E}_\theta[D] = \frac{-ke^{-\theta}}{(1 - e^{-\theta})^2} + \sum_{j \leq k}\frac{j^2 e^{-\theta j}}{(1 - e^{-\theta j})^2}.$$

Let $g_k(\theta) = \frac{-ke^{-\theta}}{(1 - e^{-\theta})^2} + \sum_{j \leq k}\frac{j^2 e^{-\theta j}}{(1 - e^{-\theta j})^2}$. By the lemma below, $g_k$ is non positive and increasing in $\theta$ for $\theta > 0$. This means we can numerically compute the inverse function of (7), with the stated error. $\qquad\square$

**Lemma G.1.** *$g_k$ is non-positive and increasing in $\theta$ for $\theta > 4\ln(2)$. Additionally, $g_k$ is decreasing in $k$.*

*Proof.* We first show non-positivity. For this, it is sufficient to show

$$\frac{j^2 e^{-\theta j}}{(1 - e^{-\theta j})^2} \leq \frac{e^{-\theta}}{(1 - e^{-\theta})^2}. \tag{12}$$

holds for all $1 \leq j$ and $\theta > 0$. This clearly holds for $j = 1$. Rearranging gives

$$j^2 e^{-\theta(j-1)} \leq \left(\frac{1 - e^{-\theta j}}{1 - e^{-\theta}}\right)^2.$$

The right-hand term is greater than or equal to 1, so it suffices to choose $\theta$ such that

$$j^2 e^{-\theta(j-1)} \leq 1$$

$$\Rightarrow \theta \geq \frac{\ln(j^2)}{j - 1}.$$

It can be shown that the right-hand term decreases with $j$ for $j > 1$, so it suffices to take $j = 2$, which implies $\theta \geq 2\ln(2)$. By choice of $\theta$ this is clearly true.

To show that $g_k$ is increasing in $\theta$, we consider

$$\frac{d}{d\theta} g_k(\theta) = \frac{ke^{-\theta}(1 - e^{-2\theta})}{(1 - e^{-\theta})^4} + \sum_{j \leq k} \frac{j^3 e^{-\theta j}(e^{-2\theta j} - 1)}{(1 - e^{-\theta j})^4}$$

Similarly, it suffices to show

$$\frac{j^3 e^{-\theta j}(1 - e^{-2\theta j})}{(1 - e^{-\theta j})^4} \leq \frac{e^{-\theta}(1 - e^{-2\theta})}{(1 - e^{-\theta})^4}.$$

Rearranging, we have

$$j^3 e^{-\theta(j-1)} \leq \frac{1 - e^{-2\theta}}{1 - e^{-2\theta j}} \left(\frac{1 - e^{-\theta j}}{1 - e^{-\theta}}\right)^4.$$

Note that

$$\frac{1 - e^{-2\theta}}{1 - e^{-2\theta j}} \left(\frac{1 - e^{-\theta j}}{1 - e^{-\theta}}\right)^4 \geq \frac{1 - e^{-\theta j}}{1 - e^{-2\theta j}}.$$

The term on the right is greater than $\frac{1}{2}$ for $\theta > 0$ and $j \geq 1$. Therefore, it suffices to choose $\theta$ such that

$$j^3 e^{-\theta(j-1)} \leq \frac{1}{2}$$

$$\Rightarrow \theta \geq \frac{\ln(2j^3)}{j - 1}.$$

Once again, the term on the right is decreasing in $j$, so we can take $j = 2$, giving $\theta \geq 4\ln(2)$, which is satisfied by our choice of $\theta$.

Finally, the fact that $g_k$ is decreasing in $k$ follows from (12). $\quad\square$

### G.3 EUCLIDEAN EMBEDDING CASE

**Theorem 4.2.** *Let $\hat{\mathbb{E}}\left[g(\lambda^a)g(y)\right]$ be an estimate of the accuracies $\mathbb{E}\left[g(\lambda^a)g(y)\right]$ using $n$ samples, where all LFs are conditionally independent given $Y$. If the signs of $a$ are recoverable, then with high probability*

$$\mathbb{E}[||\hat{\mathbb{E}}\left[g(\lambda^a)g(y)\right] - \mathbb{E}\left[g(\lambda^a)g(y)\right]||_2] = O\left(\left(a_{|min|}^{-10} + a_{|min|}^{-6}\right)\sqrt{\max(e_{max}^5, e_{max}^6)/n}\right).$$

*Here, $a_{|min|} = \min_i \left|\mathbb{E}\left[g(\lambda^i)g(y)\right]\right|$ and $e_{max} = \max_{j,k} e_{j,k}$.*

*Proof.* For three conditionally independent label functions $\lambda_a, \lambda_b, \lambda_c$, our estimate of $\mathbb{E}g(\lambda^a)g(y)]$ is

$$|\hat{\mathbb{E}}[g(\lambda^a)g(y)]| = \left( \frac{|\hat{e}_{ab}||\hat{e}_{a,c}|\mathbb{E}[Y^2]}{|\hat{e}_{b,c}|} \right)^{\frac{1}{2}}.$$

Furthermore, if we define $x_{a,b} = \frac{|\hat{e}_{a,b}|}{|e_{a,b}|}$, we can write the ratio of elements of $\hat{a}$ to $a$ as

$$k_a = \frac{|\hat{\mathbb{E}}[g(\lambda^a)g(y)]|}{|\mathbb{E}[g(\lambda^a)g(y)]|} = \left( \frac{|\hat{e}_{ij}|}{|e_{ij}|} \cdot \frac{|\hat{e}_{a,c}|}{|e_{a,c}|} \cdot \frac{|e_{b,c}|}{|\hat{e}_{b,c}|} \right)^{\frac{1}{2}} = \left( \frac{x_{a,b}x_{a,c}}{x_{b,c}} \right)^{\frac{1}{2}}.$$

(and the other definitions are symmetric for $k_b$ and $k_b$). Now note that because we assume that signs are completely recoverable, $|\hat{\mathbb{E}}[g(\lambda^a)g(y)] - \mathbb{E}[g(\lambda^a)g(y)]| = \left| |\hat{\mathbb{E}}[g(\lambda^a)g(y)]| - |\mathbb{E}[g(\lambda^a)g(y)]| \right|$.

Next note that

$$|\hat{\mathbb{E}}[g(\lambda^a)g(y)]^2 - \mathbb{E}[g(\lambda^a)g(y)]^2| = |\hat{\mathbb{E}}[g(\lambda^a)g(y)] - \mathbb{E}[g(\lambda^a)g(y)]||\hat{\mathbb{E}}[g(\lambda^a)g(y)] + \mathbb{E}[g(\lambda^a)g(y)]|.$$

By the reverse triangle inequality,

$$\begin{aligned}
(|\hat{\mathbb{E}}[g(\lambda^a)g(y)]| - |\mathbb{E}[g(\lambda^a)g(y)]|)^2 &= ||\hat{\mathbb{E}}[g(\lambda^a)g(y)]| - |\mathbb{E}[g(\lambda^a)g(y)]||^2 \\
&\leq |\hat{\mathbb{E}}[g(\lambda^a)g(y)] - \mathbb{E}[g(\lambda^a)g(y)]|^2 \\
&= \left( \frac{|\hat{\mathbb{E}}[g(\lambda^a)g(y)]^2 - \mathbb{E}[g(\lambda^a)g(y)]^2|}{|\hat{\mathbb{E}}[g(\lambda^a)g(y)] + \mathbb{E}[g(\lambda^a)g(y)]|} \right)^2 \\
&\leq \frac{1}{c^2} |\hat{\mathbb{E}}[g(\lambda^a)g(y)]^2 - \mathbb{E}[g(\lambda^a)g(y)]^2|^2,
\end{aligned}$$

where we define $c$ as $\min_a |\hat{\mathbb{E}}[g(\lambda^a)g(y)] + \mathbb{E}[g(\lambda^a)g(y)]|$. (With high probability $c = \hat{a}_{|min|} + |a_{|min|}$, but there is a chance that $\hat{\mathbb{E}}[g(\lambda^a)g(y)]$ and $\mathbb{E}[g(\lambda^a)g(y)]$ have opposite signs.) For ease of notation, suppose we examine a particular $(a, b, c) = (1, 2, 3)$. Then,

$$\begin{aligned}
&(|\mathbb{E}[g(\lambda^1)g(y)]| - |\hat{\mathbb{E}}[g(\lambda^1)g(y)]|)^2 \\
&\leq \frac{1}{c^2} |\hat{\mathbb{E}}[g(\lambda^1)g(y)]^2 - \mathbb{E}[g(\lambda^1)g(y)]|^2 = \frac{1}{c^2} \left| \frac{|\hat{e}_{12}||\hat{e}_{13}|}{|\hat{e}_{23}|} - \frac{|e_{12}||e_{13}|}{|e_{23}|} \right|^2 \\
&= \frac{1}{c^2} \left| \frac{|\hat{e}_{12}||\hat{e}_{13}|}{|\hat{e}_{23}|} - \frac{|\hat{e}_{12}||\hat{e}_{13}|}{|e_{23}|} + \frac{|\hat{e}_{12}||\hat{e}_{13}|}{|e_{23}|} - \frac{|\hat{e}_{12}||e_{13}|}{|e_{23}|} + \frac{|\hat{e}_{12}||e_{13}|}{|e_{23}|} - \frac{|e_{12}||e_{13}|}{|e_{23}|} \right|^2 \\
&\leq \frac{1}{c^2} \left( \left| \frac{\hat{e}_{12}\hat{e}_{13}}{\hat{e}_{23}e_{23}} \right| ||\hat{e}_{23}| - |e_{23}|| + \left| \frac{\hat{e}_{12}}{e_{23}} \right| ||\hat{e}_{13}| - |e_{13}|| + \left| \frac{e_{13}}{e_{23}} \right| ||\hat{e}_{12}| - |e_{12}|| \right)^2 \\
&\leq \frac{1}{c^2} \left( \left| \frac{\hat{e}_{12}\hat{e}_{13}}{\hat{e}_{23}e_{23}} \right| |\hat{e}_{23} - e_{23}| + \left| \frac{\hat{e}_{12}}{e_{23}} \right| |\hat{e}_{13} - e_{13}| + \left| \frac{e_{13}}{e_{23}} \right| |\hat{e}_{12} - e_{12}| \right)^2.
\end{aligned}$$

With high probability, all elements of $\hat{e}$ and $e$ must be less than $e_{max} = \max_{j,k} e_{j,k}$. We further know that elements of $|e|$ are at least $a_{|min|}^2/\mathbb{E}[Y^2]$. Now suppose (with high probability) that elements of $|\hat{e}|$ are at

least $\hat{a}^2_{|min|} > 0$, and define $\triangle_{a,b} = \hat{e}_{a,b} - e_{a,b}$. Then,

$$(|\mathbb{E}[g(\lambda^1)g(y)]^2| - |\mathbb{E}[g(\lambda^1)g(y)]|)^2$$

$$\leq \frac{\max(e_{max}, e^2_{max})}{c^2} \left( \frac{1}{a^2_{|min|}\hat{a}^2_{|min|}\mathbb{E}[Y^2]^2}|\triangle_{23}| + \frac{1}{a^2_{|min|}\mathbb{E}[Y^2]}|\triangle_{13}| + \frac{1}{a^2_{|min|}\mathbb{E}[Y^2]}|\triangle_{12}| \right)^2$$

$$\leq \frac{\max(e_{max}, e^2_{max})}{c^2}(\triangle^2_{23} + \triangle^2_{13} + \triangle^2_{12}) \left( \frac{1}{a^4_{|min|}\hat{a}^4_{|min|}\mathbb{E}[Y^2]^4} + \frac{2}{a^4_{|min|}\mathbb{E}[Y^2]^2} \right).$$

The original expression is now

$$|\hat{\mathbb{E}}[g(\lambda^a)g(y)] - \mathbb{E}[g(\lambda^a)g(y)]|$$

$$\leq \left( \frac{\max(e_{max}, e^2_{max})}{c^2} \left( \frac{1}{a^4_{|min|}\hat{a}^4_{|min|}\mathbb{E}[Y^2]^4} + \frac{2}{a^4_{|min|}\mathbb{E}[Y^2]^2} \right) (\triangle^2_{a,b} + \triangle^2_{a,c} + \triangle^2_{b,c}) \right)^{\frac{1}{2}}.$$

To bound the maximum absolute value between elements of $\hat{e}$ and $e$, note that

$$\left( 2 \left( \triangle^2_{ij} + \triangle^2_{a,c} + \triangle^2_{b,c} \right) \right)^{\frac{1}{2}} \leq ||\hat{e}_{a,b,c} - e_{a,b,c}||_F,$$

where $e_{a,b,c}$ is a $3 \times 3$ matrix with $e_{i,j}$ in the $(i,j)$-th position. Moreover, it is a fact that $||\hat{e}_{a,b,c} - e_{a,b,c}||_F \leq \sqrt{r}||\hat{e}_{a,b,c} - e_{a,b,c}||_2 \leq \sqrt{3}||\hat{e}_{a,b,c} - e_{a,b,c}||_2$ where $r$ is the rank of $\hat{e}_{a,b,c} - e_{a,b,c}$. Putting everything together,

$$|\hat{\mathbb{E}}[g(\lambda^a)g(y)] - \mathbb{E}[g(\lambda^a)g(y)]|$$

$$\leq \left( \frac{\max(e_{max}, e^2_{max})}{c^2} \left( \frac{1}{a^4_{|min|}\hat{a}^4_{|min|}\mathbb{E}[Y^2]^4} + \frac{2}{a^4_{|min|}\mathbb{E}[Y^2]^2} \right) \cdot \frac{1}{2}||\hat{e}_{a,b,c} - e_{a,b,c}||^2_F \right)^{\frac{1}{2}}$$

$$\leq \left( \frac{\max(e_{max}, e^2_{max})}{c^2} \left( \frac{1}{a^4_{|min|}\hat{a}^4_{|min|}\mathbb{E}[Y^2]^4} + \frac{2}{a^4_{|min|}\mathbb{E}[Y^2]^2} \right) \cdot \frac{3}{2}||\hat{e}_{a,b,c} - e_{a,b,c}||^2_2 \right)^{\frac{1}{2}}.$$

Lastly, to compute $\mathbb{E}[||\hat{\mathbb{E}}[g(\lambda^a)g(y)] - \mathbb{E}[g(\lambda^a)g(y)]||_2]$, we use Jensen's inequality (concave version, due to the square root) and linearity of expectation:

$$\mathbb{E}[|\hat{\mathbb{E}}[g(\lambda^a)g(y)] - \mathbb{E}[g(\lambda^a)g(y)]|]$$

$$\leq \left( \frac{\max(e_{max}, e^2_{max})}{c^2} \left( \frac{1}{a^4_{|min|}\hat{a}^4_{|min|}\mathbb{E}[Y^2]^4} + \frac{2}{a^4_{|min|}\mathbb{E}[Y^2]^2} \right) \cdot \frac{3}{2}\mathbb{E}[||\hat{e}_{a,b,c} - e_{a,b,c}||^2_2] \right)^{\frac{1}{2}}.$$

We use the covariance matrix inequality as described in Tropp (2014), which states that

$$P(||\hat{e} - e||_2 \geq \gamma) \leq \max \left( 2e^{3 - \frac{n\gamma}{\sigma^2 C}}, 2e^{3 - \frac{n\gamma^2}{\sigma^4 C^2}} \right),$$

where $\sigma = \max_a e_{aa}$ and $C > 0$ is a universal constant.

To get the probability distribution over $||\hat{e}_{a,b,c} - e_{a,b,c}||^2_2$, we just note that $P(||\hat{e}_{a,b,c} - e_{a,b,c}||_2 \geq \gamma) = P(||\hat{e} - e||^2_2 \geq \gamma^2)$ to get

$$P(||\hat{e}_{a,b,c} - e_{a,b,c}||^2_2 \geq \gamma) \leq 2e^3 \max \left( e^{-\frac{n\sqrt{\gamma}}{\sigma^2 C}}, e^{-\frac{n\gamma}{\sigma^4 C^2}} \right).$$

From this we can integrate to get

$$\mathbb{E}[||\hat{e}_{a,b,c} - e_{a,b,c}||_2^2] = \int_0^\infty P(||\hat{e}_{a,b,c} - e_{a,b,c}||_2^2 \geq \gamma)d\gamma \leq 2e^3 \max\left(\frac{\sigma^4 C^2}{n}, 2\frac{\sigma^4 C^2}{n^2}\right) = O(\frac{\sigma^4}{n}).$$

Substituting this back in, we get

$$\mathbb{E}[|\hat{\mathbb{E}}[g(\lambda^a)g(y)] - \mathbb{E}[g(\lambda^a)g(y)]|]$$

$$\leq \left(\frac{\max(e_{max}, e_{max}^2)}{(\hat{a}_{|min|} + a_{|min|})^2}\left(\frac{1}{a_{|min|}^4 \hat{a}_{|min|}^4 \mathbb{E}[Y^2]^4} + \frac{2}{a_{|min|}^4 \mathbb{E}[Y^2]^2}\right) \cdot O\left(\frac{\sigma^4}{n}\right)\right)^{\frac{1}{2}}$$

$$\leq \left(\frac{\max(e_{max}, e_{max}^2)}{(\hat{a}_{|min|} + a_{|min|})^2} \cdot \left(\frac{1}{a_{|min|}^4 \hat{a}_{|min|}^4 \mathbb{E}[Y^2]^4} + \frac{2}{a_{|min|}^4 \mathbb{E}[Y^2]^2}\right) \cdot O\left(\frac{\sigma^4}{n}\right)\right)^{\frac{1}{2}}$$

$$\leq \left(\frac{\max(e_{max}, e_{max}^2)}{(\hat{a}_{|min|} + a_{|min|})^2 \min(\mathbb{E}[Y^2]^4, \mathbb{E}[Y^2]^2)} \cdot \left(\frac{1}{a_{|min|}^4 \hat{a}_{|min|}^4} + \frac{2}{a_{|min|}^4}\right) \cdot O\left(\frac{\sigma^4}{n}\right)\right)^{\frac{1}{2}}$$

with high probability. Finally, we clean up the $a_{|min|}$ and $\hat{a}_{|min|}$ terms. The terms involving $a$ and $\hat{a}$ can be rewritten as

$$\frac{1 + 2\hat{a}_{|min|}^4}{a_{|min|}^6 \hat{a}_{|min|}^4 + 2a_{|min|}^5 \hat{a}_{|min|}^5 + a_{|min|}^4 \hat{a}_{|min|}^6}.$$

We suppose that $\frac{\hat{a}_{|min|}}{a_{|min|}} \in [1 - \epsilon, 1 + \epsilon]$ for some $\epsilon > 0$ with high probability. Then, this becomes less than

$$\frac{1 + 2\hat{a}_{|min|}^4}{(1-\epsilon)^4 a_{|min|}^{10} + 2(1-\epsilon)^5 a_{|min|}^{10} + (1-\epsilon)^6 a_{|min|}^{10}} \leq \frac{1 + 2\hat{a}_{|min|}^4}{4(1-\epsilon)^6 a_{|min|}^{10}}$$

$$= O\left(\frac{1}{a_{|min|}^{10}} + \frac{1}{a_{|min|}^6}\right).$$

Therefore, with high probability, the sampling error for the accuracy is bounded by

$$\mathbb{E}[||\hat{\mathbb{E}}[g(\lambda^a)g(y)] - \mathbb{E}[g(\lambda^a)g(y)]||_2] = O\left(\sigma^2\left(\frac{1}{a_{|min|}^{10}} + \frac{1}{a_{|min|}^6}\right)\sqrt{\frac{\max(e_{max}, e_{max}^2)}{n}}\right)$$

$$= O\left(\left(\frac{1}{a_{|min|}^{10}} + \frac{1}{a_{|min|}^6}\right)\sqrt{\frac{\max(e_{max}^5, e_{max}^6)}{n}}\right).$$

$\square$

Note that if all label functions are conditionally independent, we only need to know the sign of one accuracy to recover the rest. For example, if we know if $\mathbb{E}[g(\lambda^1)g(y)]$ is positive or negative, we can use $\mathbb{E}[g(\lambda^1)g(y)]\mathbb{E}[g(\lambda^2)g(y)] = e_{1,2}\mathbb{E}[Y^2]^2$, $\mathbb{E}[g(\lambda^1)g(y)]\mathbb{E}[g(\lambda^3)g(y)] = e_{1,3}\mathbb{E}[Y^2]^2, \ldots, \mathbb{E}[g(\lambda^1)g(y)]\mathbb{E}[g(\lambda^m)g(y)] = e_{1,m}\mathbb{E}[Y^2]^2$ to recover all other signs.

## H  EXTENDED EXPERIMENTAL DETAILS

In this section, we provide some additional experimental results and details. All experiments were conducted on a machine with Intel Broadwell 2.7GHz CPU and NVIDIA GK210 GPU. Each experiment takes from 30 minutes up to a few hours depending on the experiment conditions.

**Hyperbolic Space and Geodesic Regression**  The following is basic background useful for understanding our hyperbolic space models. Hyperbolic space is a Riemannian manifold. Unlike Euclidean space, it does not have a vector space structure. Therefore it is not possible to directly add points. However, geometric notions like length, distance, and angles do exist; these are obtained through the Riemannian metric for the space. Points $p$ in Smooth manifolds $M$ are equipped with a *tangent space* denoted $T_pM$. The elements (tangent vectors $v \in T_pM$) in this space are used for linear approximations of a function at a point.

The shortest paths in a Riemannian manifold are called geodesics. Each tangent vector $x \in T_pM$ is equivalent to a particular geodesic that takes $p$ to a point $q$. Specifically, the *exponential map* is the operation $\exp : T_pM \to M$ that takes $p$ to the point $q = \exp_p(v)$ that the tangent vector $v$ points to. In addition, the length $\|v\|$ of the tangent vector is also the distance $d(p, q)$ on the manifold. This operation can be reversed with the log map, which, given $q \in M$, provides the tangent vector $v = \log_p(q)$.

The geodesic regression model is the following. Set some intercept $p \in M$. Scalars $x_i \in \mathbb{R}$ are selected according to some distribution. Without noise, the output points are selected along a geodesic through $p$ parametrized by $\beta$: $y_i = \exp_p(x_i \times \beta)$, where $\beta$ is the weight vector, a tangent vector in $T_p(M)$ that is not known. This is a noiseless model; the more interesting problem allows for noise, generalizing the typical situation in linear regression. This noise is added using the following approach. For notational convenience, we write $\exp_p(v)$ as $\exp(p, v)$. Then, the noisy $y$ are $y_i = \exp(\exp(p, x_i \times \beta), \varepsilon_i)$, where $\varepsilon_i \in T_{\exp(p, x_i)}M$. This notion generalizes adding zero-mean Gaussian noise to the $y$'s in conventional linear regression.

The equivalent of least squares estimation is then given by $\hat{p}, \hat{\beta} = \arg\min_{q,\alpha} \sum_{i=1}^{n} d(\exp(q, \alpha x_i), y_i)^2$.

**Semantic Dependency Parsing**  We used datasets on Czech and English taken from the Universal Dependencies Nivre et al. (2020) repository. The labeling functions were Stanza Qi et al. (2020) pre-trained semantic dependency parsing models. These pre-trained models were trained over other datasets from the same language. For the Czech experiments, these were the models `cs.cltt`, `cs.fictree`, `cs.pdt`, while for English, they were taken from `en.gum`, `en.lines`, `en.partut`, `en.ewt`. The metric is the standard unlabeled attachment score (UAS) metric used for unlabeled dependency parsing. Finally, we used access to a subset of labels to compute expected distances, as in the label model variant in Chen et al. (2021).

**Movies dataset processing**  In order to have access to ample features for learning to rank and regression tasks, we combined IMDb (imd) and TMDb(tmd) datasets based on movie id. In the IMDb dataset, we mainly used movie metadata, which has information chiefly about the indirect index of the popularity (e.g. the number of director facebook likes) and movie (e.g. genres). The TMDb dataset gives information mainly about the movie, such as runtime and production country. For ranking and regression, we chose *vote_average* column from TMDb dataset as a target feature. In rankings, we converted the *vote_average* column into rankings so the movie with the highest review has a top ranking. We excluded the movie scores in IMDb dataset from input features since it is directly related to TMDb *vote_average*. But we later included IMDb, Rotten Tomatoes, MovieLens ratings as weak labels in Table 2.

After merging the two datasets, we performed feature engineering as follows.

1. One-hot encoding: *color, content_rating, original_language, genres, status*

2. Top (frequency) 0.5% one-hot encoding: *plot_keywords, keywords, production_companies, actors*

3. Count: *production_countries, spoken_languages*

4. Merge (add) *actor_1_facebook_likes, actor_2_facebook_likes, actor_3_facebook_likes* into *actor_facebook_likes*

5. Make a binary feature regarding whether the movie's homepage is missing or not.

6. Transform date into one-hot features such as month, the day of week.

By adding the features above, we were able to enhance performance significantly in the fully-supervised regression task used as a baseline.

In the ranking experiments, we randomly sampled 5000 sets of movies as the training set, and 1000 sets of movies as the test set. The number of items in an item set (ranking set) was 5. Note that the movies in the training set and test set are strictly separated, i.e. if a movie appears in the training set, it is not included in the test set.

**Model and hyperparameters**   In the ranking setup, we used 4-layer MLP with ReLU activations. Each hidden layer had 30 units and batch normalization(Ioffe & Szegedy, 2015) was applied for all hidden layers. We used the SGD optimizer with ListMLE loss (Xia et al., 2008); the learning rate was 0.01.

In the regression experiments, we used gradient boosting regression implemented in sklearn with *n_estimators*=250. Other than *n_estimators*, we used the default hyperparameters in sklearn's implementation.

**BoardGameGeek data processing**   In the BoardGameGeek dataset (boa, 2017), we used metadata of board games only. Since this dataset showed enough model performance without additional feature engineering, we used the existing features: *yearpublished, minplayers, maxplayers, playingtime, minplaytime, maxplaytime, minage, owned, trading, wanting, wishing, numcomments, numweights, averageweight*. The target variable was average ratings (*average*) for regression, and board game ranking (*Board Game Rank*). Note that 'Board Game Rank' cannot be directly calculated from average ratings. BoardGameGeek has its own internal formula to determine the ranking of board games. In the ranking setup, we randomly sampled 5000 board game sets as the training set, and 1000 board game sets as the test set.

**Simulated LFs generation in ranking and regression**   In rankings, we sampled from $\lambda^a(i) \sim \frac{1}{Z}e^{-\beta_a d_\tau(\pi, Y_i)}$ with $\beta_a$. For more realistic heterogeneous LFs, 1/3 (good) LFs $\beta_a$ are sampled from $Uniform(0.2, 1)$, and 2/3 (bad) LFs' $\beta_a$ are sampled from $Uniform(0.001, 0.01)$. Note that higher value of $\beta_a$ yields less noisy weak labels. In regression, we used the conditional distribution $\Lambda|Y$ to generate samples of $\Lambda$. Specifically, where $\Lambda|Y \sim \mathcal{N}(\bar{\mu}, \bar{\Sigma})$, where $\bar{\mu} = \Sigma_{\Lambda Y}\Sigma_Y^{-1}y$ and $\bar{\Sigma} = \Sigma_\Lambda - \Sigma_{\Lambda Y}\Sigma_Y^{-1}\Sigma_{Y\Lambda}$, from the assumption $(\Lambda, Y) \sim \mathcal{N}(0, \Sigma)$.

# I   ADDITIONAL SYNTHETIC EXPERIMENTS

We present several additional synthetic experiments, including results for partial ranking labeling functions and for parameter recovery in regression settings.

## I.1   RANKING

To check whether our algorithm works well under different conditions, we performed additional experiments with varied parameters. In addition, we performed a partial ranking experiment.

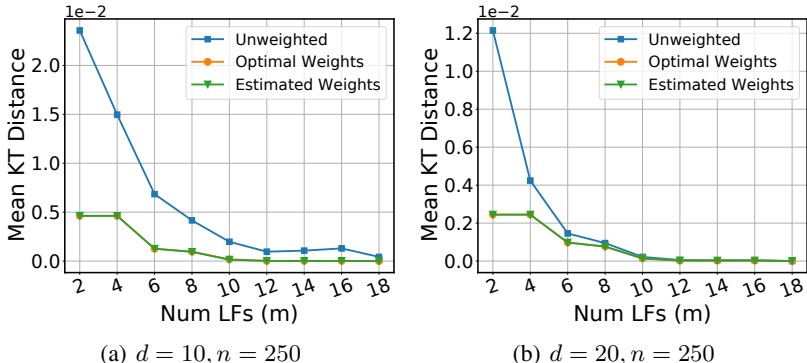

(a) $d = 10, n = 250$             (b) $d = 20, n = 250$

Figure 6: Inference via weighted and standard Kemeny rule over full rankings (top) with permutations of size $d = 10, 20$. Error metric is Kendall tau distance (lower is better). Proposed weighted Kemeny rule is nearly optimal on full rankings.

**Ranking synthetic data generation** First, $n$ true rankings $Y$ are sampled uniformly at random. In the full ranking setup, each LF $\pi_i$ is sampled from the Mallows $\lambda^a(i) \sim \frac{1}{Z} e^{-\beta_a d_\tau(\pi, Y_i)}$ with parameters $\beta_a, Y_i$ and in the partial ranking setup it is sampled from a selective Mallows with parameters $\beta_a, Y_i, S_i$ where each $S_i \subseteq [d]$ is chosen randomly while ensuring that each $x \in [d]$ appears in at least a $p$ fraction of these subsets. Higher $p$ corresponds to dense partial rankings and smaller $p$ leads to sparse partial rankings. We generate 18 LFs with 10 of then having $\beta_a \sim Uniform(0.1, 0.2)$ and rest have $\beta_a \sim Uniform(2, 5)$. This was done in order to model LFs of different quality. These LFs are then randomly shuffled so that the order in which they are added is not a factor. For the partial rankings setup we use the same process to get $\beta_a$ and randomly generate $S_i$ according to the sparsity parameter $p$. For a set of LFs parameters we run the experiments for 5 random trials and record the mean and standard deviation.

**Full Ranking Experiments** Figure 6 shows synthetic data results without an end model, i.e., just using the inference procedure as an estimate of the label. We report the 'Unweighted' Kemeny baseline that ignores differing accuracies. 'Estimated Weights' uses our approach, while 'Optimal Weights' is based on an oracle with access to the true $\theta_a$ parameters. As expected, synthesis with estimated parameters improves on the standard Kemeny baseline. The improvement for the full rankings case (top) is higher for *fewer* LFs; this is intuitive, as adding more LFs globally improves estimation even when accuracies differ.

**Partial Ranking Experiments** In the synthetic data partial ranking setup, we vary the value of $p$ (observation probability) from 0.9 (dense) to 0.2 (sparse) and apply our extension of the inference method to partial rankings. Figure 7 shows the results obtained. Our observations in terms of unweighted vs weighted aggregation remain consistent here with the full rankings setup. This suggests that the universal approach can provide the same type of gains in the partial rankings.

### I.2 REGRESSION

Similarly, we performed a synthetic experiment to show how our algorithm performs in parameter recovery.

**Regression synthetic data generation** The data generation model is linear $Y = \beta^T X$, where our data is given by $(X, Y)$ with $X \in \mathbb{R}^q$ and $Y \in \mathbb{R}$. We generate $n$ such samples. Note that we did not add a noise variable $\varepsilon \sim \mathcal{N}(0, \sigma^2)$ here since we do not directly interact with $Y$; the noise exists instead in the labeling functions (i.e., the weak labels).

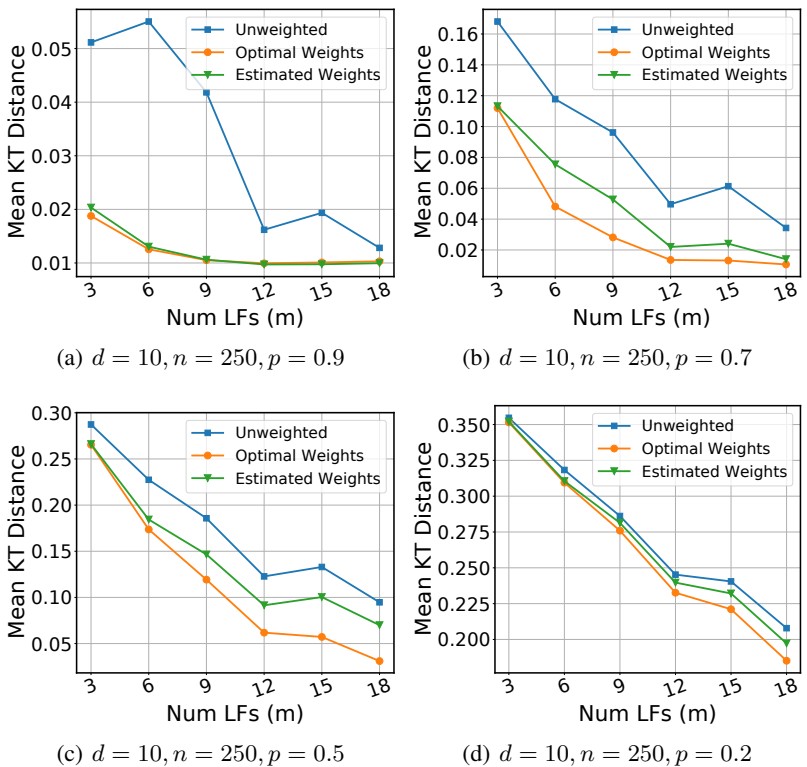

Figure 7: Inference via weighted and standard majority vote over partial rankings with permutations of size $d = 10$. Error metric is Kendall tau distance (lower is better). Proposed inference rule is nearly optimal on full rankings.

**Parameter estimation in synthetic regression experiments** Figure 8 reports results on synthetic data capturing label model estimation error for the accuracy and correlation parameters $(\mu, \sigma)$ and for directly estimating the label $Y$. As expected, estimation improves as the number of samples increases. The top-right plot is particularly intuitive: here, our improved inference procedure vastly improves over naive averaging as it accesses sufficiently many samples to estimate the label model itself. On the bottom, we observe, as expected, that label estimation significantly improves with access to more labels.

# J  ADDITIONAL REAL LF EXPERIMENTS AND RESULTS

We present a few more experiments with different types of labeling functions.

## J.1  BOARD GAME GEEK DATASET

In the board games dataset, we built labeling functions using simple programs expressing heuristics. For regression, we picked several continuous features and scaled them to a range and removed outliers. Similarly, for rankings, we picked what we expected to be meaningful features and produced rankings based on them. The selected features were ['owned', 'trading', 'wanting', 'wishing', 'numcomments'].

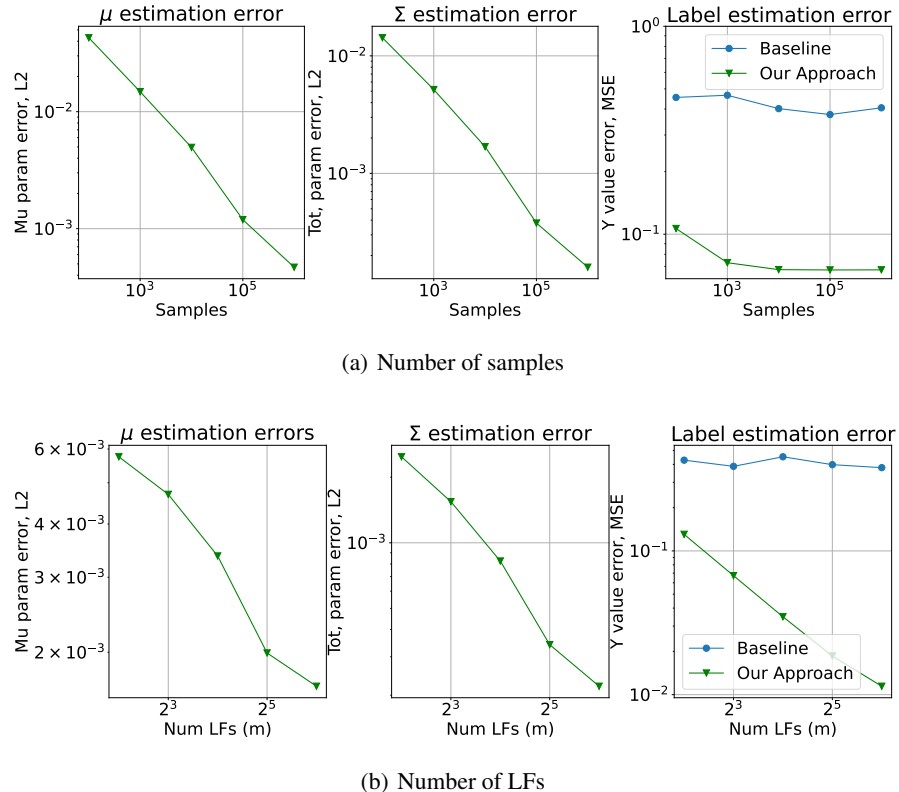

(a) Number of samples

(b) Number of LFs

Figure 8: Parameter and label estimation with varying the number of samples (top) and the number of labeling functions (bottom)

|  | # of training examples | Kendall Tau distance (mean $\pm$ std) |
|---|---|---|
| Fully supervised (5%) | 250 | $0.1921 \pm 0.0094$ |
| Fully supervised (10%) | 500 | $0.1829 \pm 0.0068$ |
| WS (Heuristics) | 5000 | $0.1915 \pm 0.0011$ |

Table 5: End model performance with true ranking LFs in BGG dataset.

In this case, we observed that despite not having access to any labels, we can produce performance similar to fully-supervised training on 5-10% of the true labels. We expect that further LF development will produce even better performance.

## J.2 MSLR-WEB10K

To further illustrate the strength of our approach we ran an experiment using unsupervised learning methods in information retrieval (such as BM25) as weak supervision sources. The task is information retrieval, the dataset is MSLR-WEB10Kmsl, and the model and training details are identical to the other experiments. We used several labeling functions including BM25 and relied on our framework to integrate these. The labeling functions were written over BM25 and features such as covered query term number, covered query

|  | # of training examples | MSE (mean $\pm$ std) |
|---|---|---|
| Fully supervised (1%) | 144 | $0.4605 \pm 0.0438$ |
| Fully supervised with "bad" subset (10%) | 1442 | $0.8043 \pm 0.0013$ |
| Fully supervised with "bad" subset (25%) | 3605 | $0.4628 \pm 0.0006$ |
| WS (Heuristics) | 14422 | $0.8824 \pm 0.0005$ |

Table 6: End model performance with true regression LFs in BGG dataset. The training data was picked based on the residuals in linear regression (resulting in a "bad" subset scenario for a challenging dataset). We obtain comparable performance.

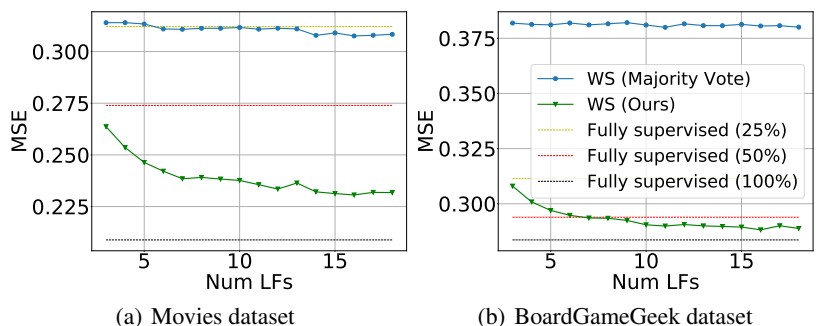

      (a) Movies dataset        (b) BoardGameGeek dataset

Figure 9: End model performance with regression LFs (Left: Movies dataset, Right: BGG Dataset). Results from training a model on pseudolabels are compared to fully-supervised baselines on varying proportions of the dataset. Baseline is the averaging of weak labels. Metric is (MSE); lower is better.

term ratio, boolean model, vector space model, LMIR.ABS, LMIR.DIR, LMIR.JM, and query-url click count.

As expected, the synthesis of multiple sources produced better performance than BM25 Dehghani et al. (2017) alone. Despite not using any labels, we outperform training on 10% of the data with true labels. This suggests that our framework for integrating multiple sources is a better choice than either hand-labeling or using a single source of weak supervision to provide weak labels. Below, for Kendall tau, lower is better.

|  | Kendall tau distance | NDCG@1 |
|---|---|---|
| Fully supervised (10%) | $0.4003 \pm 0.0151$ | $0.7000 \pm 0.0200$ |
| Fully supervised (25%) | $0.3736 \pm 0.0090$ | $0.7288 \pm 0.0077$ |
| WS (Dehghani et al. (2017)) | $0.4001 \pm 0.0063$ | $0.7288 \pm 0.0077$ |
| WS (Ours) | $0.3929 \pm 0.0052$ | $0.7402 \pm 0.0119$ |

Table 7: End model performance with true ranking LFs in MSLR-WEB10K dataset. Since the dataset has a lot of tie scores and the number of items is not uniform across examples, we sampled the examples with five unique scores (0, 1, 2, 3, 4). Also, in each example, items are randomly chosen so that each score occurs only once in each item set.

