# OpenReview forum: "Universalizing Weak Supervision"
_ICLR.cc/2022/Conference — ICLR 2022 Poster_

### Official Review · Reviewer_NZTU · 2021-11-02

**Correctness:** 4
**Technical Novelty And Significance:** 3
**Empirical Novelty And Significance:** 3
**Recommendation:** 8
**Confidence:** 3

**Main Review:**

A new general framework for weak supervision is proposed. The main idea is to learn embeddings over the labels tractably into two embedding spaces (Boolean/Euclidean). Then, estimators for the parameters of the labeling model is learned using MLE. It is proved that if the embeddings are isometric the estimators are consistent and convergence rates are proven for these estimators. Experiments are performed on varied tasks and the results compare the proposed approach with Snorkel to show improved performance in weakly supervised learning for these tasks.

The paper is well written and has a very general framework for weak supervision. The experiments show the benefits of this framework in varied tasks. Also, there are new problems that the proposed framework is applied to (not previously possible by other weakly supervised approaches). The paper seems well written, has both good experiments and theoretical guarantees. One minor point is perhaps there is not a lot in discussion in terms of prior work for someone who is not familiar with the specific area. Is snorkel the only possible comparison baseline (particularly since a lot of the tasks are novel and cannot be done by snorkel).


**Summary Of The Paper:**

A novel general approach for weakly supervised learning.
+ A new model for incorporating weak sources based on an exponential family parameterization
+ Theoretical guarantees on the estimators for learning the parameterization
+ Empirical results show that the proposed approach can apply weak supervision to novel problem types

**Summary Of The Review:**

Overall, this seems like a strong paper with good theoretical justification and experimental results. The one possible weakness is perhaps lack of comparison baselines (though not sure if it is possible or not)

---

> ### Author Response · Authors · 2021-11-18
> **Response to NZTU**
>
> We sincerely thank the reviewer for their kind words, constructive feedback, and suggestions. The reviewer appreciated the generality of our work and the variety of applications in enables.
>
> * **On the comparison baselines**: The reviewer is exactly right here: the problems we tackle do not have existing approaches in weak supervision, beyond the definition of majority vote, which was also generalized to the metric space setting. However, we did seek to use Snorkel as a baseline in the one setting where it was possible: the rankings case. Snorkel does not apply to any continuous setting. For rankings, Snorkel treats each permutation as a different class in multiclass classification. That is, it completely ignores the structure given by the distance between permutations. As a result, it is not even as good as majority vote, which does take the distance (but not LF quality differences) into account. Our technique takes **both into account** and does so for a huge number of potential applications.

---

### Official Review · Reviewer_NJ4o · 2021-11-02

**Correctness:** 3
**Technical Novelty And Significance:** 3
**Empirical Novelty And Significance:** 3
**Recommendation:** 3
**Confidence:** 4

**Main Review:**

I commend the authors on their thoroughness and the aims of their approach - which are much needed in the field of weak supervision. I find some of the potential scenarios really interesting and appealing, particularly learning in hyperbolic space.

My concerns are threefold. The first has to do with novelty. There are many approaches in the past that also move past classification to consider ranking and regression. In general in loss factorizations of the form of

Loss factorization, weakly supervised learning and label noise robustness
Giorgio Patrini, Frank Nielsen, Richard Nock, Marcello Carioni Proceedings of The 33rd International Conference on Machine Learning, PMLR 48:708-717, 2016.

that apply to most of those settings. In the following, the authors can also find the explicit generalization including classification, ranking or regression, with a similar notion of pseudolabels:

Cid-Sueiro J., García-García D., Santos-Rodríguez R. (2014) Consistency of Losses for Learning from Weak Labels. In: Calders T., Esposito F., Hüllermeier E., Meo R. (eds) Machine Learning and Knowledge Discovery in Databases. ECML PKDD 2014. Lecture Notes in Computer Science, vol 8724. Springer, Berlin, Heidelberg. https://doi.org/10.1007/978-3-662-44848-9_13

And even further generalizations from weak to superset learning, which arguably could be seen as including the former.

Hüllermeier E., Cheng W. (2015) Superset Learning Based on Generalized Loss Minimization. In: Appice A., Rodrigues P., Santos Costa V., Gama J., Jorge A., Soares C. (eds) Machine Learning and Knowledge Discovery in Databases. ECML PKDD 2015. Lecture Notes in Computer Science, vol 9285. Springer, Cham. https://doi.org/10.1007/978-3-319-23525-7_16

It is not clear to me how these overlap with this work. Also, the structure of the paper does not help in understanding this, with the related work being in the supplementary material. Finally, the title seems a bit to general for what it is shown in the paper.

**Summary Of The Paper:**

The authors present a framework that intends to extend the label types considered traditionally in learning from weak labels.

**Summary Of The Review:**

In its current form, the structure of the paper makes it difficult to assess the novelty. Also, there are previous approaches in the literature that seem to overlap with the aim of the paper.

---

> ### Author Response · Authors · 2021-11-18
> **Response to NJ4o**
>
> We thank the reviewer for the their comments and references.
>
> * **On "weak supervision" terminology**. The reviewer asks about novelty and related work on weak supervision. The term **weak supervision is overloaded**. As we explain in our abstract, introduction, and related work, we are referring specifically to a class of techniques that take multiple noisy sources for each label and then synthesize them into a single pseudolabel. These pseudolabels can then be used to train any suitable downstream model. This is a **two-stage technique**, consisting of learning and using a label model for synthesis and then training one or more end models for downstream training. **Our contribution is to significantly generalize the label model**. An excellent overview of the weak supervision techniques we are concerned with is found in the recent benchmark paper [4].
>
> * **On our contribution**. The two-stage weak supervision framework described above has become very popular and succesful. The major limitation, however, is that so far, each problem setting requires developing a new label model. This has limited the application of this framework to a few problem settings: binary or multiclass classification, or simple sequence problems. **We overcome this bottleneck** by proposing an algorithm that will operate with labels in any metric space. This permits us to **tackle many kinds of problems**: as sample applications, we consider rankings, regression, Riemannian manifolds, graphs, and dependency parse trees, though our approach is fully general. Our goal is **not to develop new end models** for these settings, nor end models that deal with noisy data. Instead, our work enables weak supervision to provide pseudolabels that can then be used to train such models. This flexibility of not being locked into any particular end model is the reason behind the success of the weak supervision frameworks we generalize.
>
> * **On suggested references**. The papers [1,2,3] described by the reviewer operate on a different meaning of "weak supervision." For example, [1] considers noisy or missing labels (as opposed to multiple noisy sources) and provides a modified loss function that directly learns a model over the noisy labels. This is a **different problem setting**. Indeed, it is not competitive, but potentially cooperative with our approach, since practitioners could use the approach in [1] to train an end model on the pseudolabels we generate. Similarly, [2] proposes an analysis of end model loss functions. Finally, [3] proposes a method for training end models on multiple labels where the true label is among the sources, which is a strong assumption that the vast majority of modern weak supervision methods, including ours, do not make. Summarizing this, the key difference compared to our work, **the cited works propose new end models, while we propose a new label model** compatible with any suitable end model.
>
> [1]: Loss factorization, weakly supervised learning and label noise robustness Giorgio Patrini, Frank Nielsen, Richard Nock, Marcello Carioni Proceedings of The 33rd International Conference on Machine Learning, PMLR 48:708-717, 2016.
>
> [2]: Cid-Sueiro J., García-García D., Santos-Rodríguez R. (2014) Consistency of Losses for Learning from Weak Labels. In: Calders T., Esposito F., Hüllermeier E., Meo R. (eds) Machine Learning and Knowledge Discovery in Databases. ECML PKDD 2014. Lecture Notes in Computer Science, vol 8724. Springer, Berlin, Heidelberg. https://doi.org/10.1007/978-3-662-44848-9_13
>
> [3]: Hüllermeier E., Cheng W. (2015) Superset Learning Based on Generalized Loss Minimization. In: Appice A., Rodrigues P., Santos Costa V., Gama J., Jorge A., Soares C.
>
> [4] Zhang J., Yu Y.,  Li Y., Wang Y., Yang Y., Yang M., Ratner A. WRENCH: A Comprehensive Benchmark for Weak Supervision. https://openreview.net/pdf?id=Q9SKS5k8io

---

### Official Review · Reviewer_AswL · 2021-11-03

**Correctness:** 4
**Technical Novelty And Significance:** 3
**Empirical Novelty And Significance:** 2
**Recommendation:** 8
**Confidence:** 2

**Main Review:**

Strength
* The proposed embedding-based approximate estimation approach is novel.
* Theoretical bounds are derived to monitor the closeness of this approximate estimation.
* Experiments are performed in multiple domains with different types of labels

Weakness
1. Writing can be substantially improved.

 A lot of context is missing in the main paper, which makes the paper very hard to follow. Specifically

* Needs a bit more background introduction. "LF" (label function) in section 1 is defined in section 2.
* How θa and θab are inferred from accuracies and correlations is not clear. Seems to be based on results from prior works. As this is the key innovation, would of great help to include a illustration of the construction of algorithms 1 and 3.
* The meaning of correlation set E in equation (1) is undefined.
* I find the advantage of a learned model over majority vote straightforward and well-studied, and can be compressed to include more background discussions.

 Experiments are lousily organized. Readers have to go to the appendix for a more comprehensive picture of the proposed approach.

* Experiments in Figure 1,2 and 3 are repetitive and can be summarized.
* A selection of experiments on estimation quality (figure 5, 6 and 7) would strengthen the main paper.

2. Discussions and experiments on the universality aspect is lacking. This work claims to be a universal approach for WS, which boils down to an embedding-based approximation of distance function d(,). It is a question whether such an approximation could be made close enough to a usable level for truly universal application. It would be nice to remark at Theorem 4.2 on how accurate the embedding approximations have to be for θ to be accurate using a practical example. Is being able to tractably estimate θ but not being loyal to d(,) a superior outcome than using a not-so-close estimate of θ?

-----------------------------------------

Update: My concerns on writing and universality have been sufficiently addressed with the latest revision.

**Summary Of The Paper:**

This work studies a weakly supervised learning setup of aggregating multiple weak sources of labels into high-quality pseudo labels for learning.

The proposed approach is based on building graphical models with the true labels as latent variables following existing techniques, but generalizes existing works in 1) supporting more types of task labels such as ranking, regression and in non-Euclidean spaces through approximating the graphical model terms using dot product between embeddings, and 2) algorithms for efficient learning under the embedding approximation and theoretical bounds on the estimation error using embedding-based approximation.

Experiments focus on comparing weakly supervised learning performance of learning a graphical model versus using majority vote on the newly supported tasks and fully supervised learning with varying amount of labels. Results show that learning the graphical model achieved better performance than majority vote and can achieve parity reach a certain level of fully supervision.

**Summary Of The Review:**

My recommendation of this paper is based on the novelty with respect to previous works along this line of work, also considering the writing and experiment quality. I am not a domain expert and I didn't verify the correctness of equations and proofs.

My main concerns are 1) the proposed approach is not explained clearly (or am I missing the context?) and 2) the experiments included in the main paper seem shallow and repetitive.

---

> ### Author Response · Authors · 2021-11-18
> **Response to AswL**
>
> We appreciate the constructive feedback! It has significantly improved our updated draft.
>
> * **On writing**. We agree and have sought to add clarity in our updated paper. We have added additional details, taken care to clarify terminology before use (we thank you for spotting our use of "LF"), and added a figure (Figure 1 in our updated draft) that illustrates examples and shows a pipeline that involves the stages of our label model technique and their inputs and outputs.
>
> * **On experiments**. We agree that in our submitted draft, Figures 1,2, and 3 make the same point. As the reviewer suggests, we have replaced one of them with a parameter and label estimation experiment. These show that in the synthetic setting where the true parameters are known, the estimation error drops in the number of samples; this confirms the theoretical results in Theorem 2.
>
> * **On universality**. Indeed, this is one the most interesting aspects of our work. The reviewer asks whether consistent estimation in the embedding space is worth the price of incurring the distortion of the embedding. To address this question, we have added an additional theoretical result (Theorem 3 in our updated draft). We sketch out the idea below: essentially, the embedding method will distort the mean parameters, i.e., $E[d(\lambda^a, \lambda^b)]$. If we stuff all of the mean parameters into a single mean parameter vector $\mu$, then we can think of the distortion as controlling an upper bound on the norm $\|\mu - \mu'\|$, where $\mu$ is the true set of parameters and $\mu'$ is the distorted counterpart. Then, we show that we can control this difference as $\|\mu\|\varepsilon$, where the (multiplicative) distortion is $1-\varepsilon$. Moreover, it is possible to show that the canonical parameters, which we write as $\theta$ and $\theta'$, respectively, satisfy $\|\theta - \theta'\| \leq c \|\mu - \mu'\| \leq c\|\mu\|\varepsilon$. This is exciting: it means **the estimation error goes to zero linearly in the distortion**. Even in the worst-case, we can bound the distortion as a small constant; but in most cases of interest, there are embeddings that can make distortion arbitrarily low, at the potential cost of more dimensions, which our technique is not bothered by. In addition, the distortion term (the $\varepsilon$) can be verified directly by the user. The result we have provided opens up an exciting area---studying the theoretical connection between parameter estimation in graphical models (our work) and theoretical computer science questions on the quality of embeddings of metric spaces into $\ell_2$ and other spaces.

---

### Official Review · Reviewer_xXw5 · 2021-11-09

**Correctness:** 4
**Technical Novelty And Significance:** 2
**Empirical Novelty And Significance:** 3
**Recommendation:** 5
**Confidence:** 3

**Details Of Ethics Concerns:**

The authors bring up concerns related to bias for the weak supervision domain itself. I believe it is not a

**Main Review:**

Strengths:

- The paper is clear and well written with proper notations.
- The authors emphasize an important aspect of weak supervision (allowing different types of labels and unifying.)

Weaknesses:

- I find the theoretical contributions on the more trivial side. Finding the roots of the triplets methods is just plain algebra and error term easily follows from bernoulli assumption.
- The dataset is constructed by the authors, so it is hard to evaluate its performance. The criteria metrics might not be fair since they are directly related to this paper's distance choices.
- Since the paper claims to be a universal framework, I would expect using different types of labels in the same task (like if we have binary labels for some data and probabilistic outputs for others), or coexisting more than one task type is doable. However, in their formulation it is not, since the distances would need to be mapped on the same space or at least a similar range.
- I believe Related Work section is significiant and should be in the main paper, not in the supplementary.


Questions:
- How do you compare your dataset with [1]?
- How do you compare method for varying  label density(low, middle, high) vs majority voting? Why snorkel works worse than majority voting in your experiments?
- How would you adapt your method for the problems that use different label types at the same time? (rather than choosing one type of distance based on the problem)



[1] Zhang, Jieyu, et al. "WRENCH: A Comprehensive Benchmark for Weak Supervision." arXiv preprint arXiv:2109.11377 (2021).

**Summary Of The Paper:**

The paper proposes a framework for weak supervision which works for both discrete and continuous labels. The proposed framework assumes the sources are coming from an exponential family distribution and generalizes the accuracy and correlation terms in previous approaches. They use a discriminative learning setting and use triplet methods in order to compute accuracies. They evaluated the performance for regression, ranking and hyperbolic and graph learning tasks.

The paper's main contribution is putting an exponential family distribution assumption which allows to use any metric space. In their experiments, they choose distances based on the problem and label type, mostly weighted L2 distance except hyperbolic dataset.






**Summary Of The Review:**

In general, the paper is written clearly, connects previous work in a good way, so it is valuable in that sense. I am willing to chance my vote towards accept if the authors could clarify my concerns and convince me their work has sufficient contributions to be published.

---

> ### Author Response · Authors · 2021-11-18
> **Response to xXw5 (2/2)**
>
>
> Table 2 Ranking - additional evaluation on metrics: NDCG@1, NDCG@3, NDCG@5 (higher is better)
>
> |             |$\mathbf{d_{\tau}(y, \hat{y})}$**| NDCG@1 | NDCG@3  | NDCG@5  |
> |-------               | :--------------------- | :-----:| :-----: | :-----: |
> |Fully supervised (10%)| 0.2731                 | 0.7559 | 0.8225  | 0.9078  |
> |Fully supervised (25%)| 0.2465                 | 0.7895 | 0.8456  | 0.9198  |
> |Fully supervised (50%)| 0.2313                 | 0.8073 | 0.8592  | 0.9269  |
> |Fully supervised (100%)| 0.2282                | 0.8145 | 0.8602  | 0.9283  |
> |WS (One LF, Rotten tomatoes) | 0.2495          | 0.7932 | 0.8440  | 0.9201  |
> |WS (One LF, IMDb score) | 0.2289               | 0.8082 | 0.8591  | 0.9271  | |WS (One LF, MovieLens score) | 0.2358          | 0.8037 | 0.8553  | 0.9251  |
> |WS (3 LFs, MV) | **0.2273**                    | **0.8135** | **0.8611**  | **0.9284**  |
> |WS (3 LFs, Ours) | **0.2274**                  | **0.8140** | **0.8608**  | **0.9286**  |
> |WS (3 scores + 3 bad LFs, MV) | 0.2504         | 0.7935 | 0.8413  | 0.9198  |
> |WS (3 scores + 3 bad LFs, Ours)  | **0.2437**  | **0.7977** | **0.8450**  | **0.9205**  |
>
>
> Table 2 Regression - additional evaluation on metrics: MAE (lower is better)
>
> |                                 |            MSE       |         MAE       |
> |-------                          | :------------------: |           :-----: |
> |Fully supervised (10%)           |        0.3357        |          0.449    |
> |Fully supervised (25%)           |        0.2705        |       0.3970      |
> |Fully supervised (50%)           |        0.2399        |       0.3721      |
> |Fully supervised (100%)          |        0.2106        |       0.3511      |
> |WS (One LF, Rotten tomatoes)     |        0.4272        |       0.4374      |
> |WS (One LF, IMDb score)          |        0.2990        |       0.4043      |
> |WS (One LF, MovieLens score)     |        0.2690        |       0.5125      |
> |WS (3 LFs, MV)                   |        0.2754        |       0.4080      |
> |WS (3 LFs, Ours)                 |        **0.2451**    |    **0.3809**     |
>
> Figure 3 Geodesic regression - additional report on metrics: absolute distance (lower is better)
>
> |                                 |          Absolute distance |
> |-------                          | :------------------:       |
> |Fully supervised (80%)           |        8.9218              |
> |Fully supervised (90%)           |        8.9022              |
> |Fully supervised (100%)          |        8.9011              |
>
> |       # of LFs                  |           WS (MV)    |    WS (Ours)  |
> |-------                          | :------------------: |       :-----: |
> |3|8.9001|8.8587|
> |6|8.8478|8.8456|
> |9|8.8485|8.8434|
> |12|8.8440|8.8395|
> |15|8.8554|8.8502|
> |18|8.8478|8.8432|
> |21|8.8419|8.8408|

---

> ### Author Response · Authors · 2021-11-18
> **Response to xXw5 (1/2)**
>
> The reviewer enjoyed our work and had several questions and comments. We appreciate the suggestion on mixed label spaces and have updated our paper to include an extension to this setting. We answer each question in detail below.
>
> * **On our contribution**: The reviewer notes that our choice of exponential family model enables us to work with any metric space---**a major generalization** of previous weak supervision works. The distances we work with in our sample applications are varied: the Kendall-tau distance for permutations, the $\ell_2$ distance, the hyperbolic distance in the hyperboloid model, the graph geodesic (shortest-path) distance for graph applications, and more. Indeed, our embeddings-based approach enables us to handle highly diverse distances as needed for any task. This is important: weak supervision has become extremely popular and successful, but, prior to our work, cannot be applied to many problem types beyond discrete classification and sequence tagging. Our work **resolves this bottleneck; it extends the benefits of weak supervision to numerous settings.**
>
> * **On datasets**: Our updated work consists of a mixture of different datasets, some of which we built and will release to the public and some that are standard (i.e., the semantic dependency parsing application uses the Universal Dependencies datasets, which are canonical for this task). The **key distinction** compared to the datasets in the Wrench benchmark is that Wrench only uses label models that handle discrete classification and sequence tasks, and its datasets reflect this. Our universal approach goes beyond these tasks to be able to handle rankings, regression, learning on manifolds, parse trees, etc. The label models studied in Wrench do not apply to these settings. When we use our universal approach in the Wrench setting, our algorithm specializes to the label models used in the Wrench benchmark.
>
> * **On the simplicity of the algorithm**. Indeed, as the reviewer notes, our algorithm requires simple operations---solving linear systems or solving quadratic equations. This is a **strength of our approach**---we need not perform complex iterative optimization. The theoretical analysis, however, is quite challenging and requires careful control of a variety of error terms. The takeaway is that our approach is **both practical and offers theoretical guarantees**.
>
> * **On choices of metrics**. Based on the goal task, the practitioner chooses the weak supervision model (equation (1) in our submission), so it is natural to match the distance in this model with the task metric. However, it is not always possible to match the task metric; our approach is **robust to differing evaluation metrics**. For example, in the dependency parsing example, we use an $\ell_2$ norm-based distance on adjacency matrices, while the evaluation metric is the unlabeled attachment score (UAS). Below, we also show some examples of the results in the paper when we choose a different evaluation metric (table at bottom). For example, we run the label model with the Kendall-tau distance, but then evaluate using the nDCG metric.
>
> * **On Snorkel's weaknesses**. The reviewer noted that Snorkel is outperformed by majority vote, which is outperformed by our approach when LF quality differs. The reason for this is the following: **Snorkel ignores structure in the label metric space**, viewing each value of the label as a different class in a multiclass setting. So, for example, in the ranking setting, nearby permutations (differing by say, one swap of a single adjacent pair of elements) are treated as being as distant as any pair of permutations. Majority vote takes the structure into account by including distances in its estimate; however, it ignores differing LF quality. Our approach **takes both---label space structure and LF quality---into account**.
>
> * **On the aggregation of different types of labels**: We thank the reviewer for this suggestion. Our technique easily extends to this case. We have added an explanation of how to do so in the updated paper's Appendix C. We sketch the idea below. We can construct a product space of possible label types. So, suppose the labels can come from $\mathcal{Y}_1$, $\mathcal{Y}_2$, and so on. We can then construct a product metric space $\mathcal{Y}_1 \times \mathcal{Y}_2 \times \cdots$ whose metric is the combined metric of the factor spaces. Then our approach reduces to the factors, and inference simply operates on the combined distance.

---

### Author Response · Authors · 2021-11-18
**Common response to all authors**

We would like to thank all of the reviewers for their insightful comments and feedback. Before diving into our in-depth responses, we highlight a set of post-submission improvements in our updated draft.

* We have added a **further application** of our approach---semantic dependency parsing. Now, our label space is the space of trees. We have found that our technique is capable of improving on strong baseline parsers. With this addition, we have five diverse applications that could not be tackled with weak supervision frameworks like Snorkel (rankings, linear regression, regression on manifolds, graph estimation, dependency parsing). We are excited to see further applications that our general framework enables.

* We have added a **theoretical result** that captures the penalty of distortion (due to the embedding function) on the canonical parameter estimation error. Specifically, the error is linear in the distortion coefficient. This means that our approach is useful in the broad variety of settings where low-distortion embeddings are available. Better yet, there are available upper bounds on distortion for any metric space, so that our estimation error is bounded in the worst case.

* We have added detail to our draft to clarify our approach, including an easy-to-interpret visual (Figure 1 in our updated draft) that shows examples of applications along with a pipeline for our proposed framework.

* We have also included a **simple extension of our framework that permits the usage of multiple label types**, such as combinations of different kinds of labels. In brief, the extension works by simply constructing a product metric space whose factors are the specific label spaces being combined. Our framework then reduces to performing label model estimation across the individual factors.

We also wish to briefly reiterate the strengths  of our work and its contribution. Specifically,

* We **significantly generalize popular and successful weak supervision frameworks** to handle a variety of tasks. All that is required is for the labels to live in a metric space, a sufficiently general setting for a huge range of possible tasks.

* We introduce **efficient and simple** approaches to synthesize labels in these spaces. Our framework combines an embedding with a method-of-moments closed-form estimator. In addition, our approach comes with **theoretical guarantees**. For isometric embeddings, our estimator is consistent; we provide sample complexities. For non-isometric embeddings, our results bound the inconsistency in terms of the distortion.

* We demonstrate our framework on a **variety of applications**. We include a diverse set of five applications that weak supervision frameworks prior to ours could not handle.

---

> ### Author Response · Authors · 2021-11-26
> **Thank you again; further questions?**
>
> Dear Reviewers,
>
> We want to thank you again for your feedback, questions, and suggestions. We believe we have answered all of your questions in our responses and the updated draft.
>
> If you have further questions, we would love to answer them.

---

### Decision · Program_Chairs · 2022-01-20

**Decision:**

Accept (Poster)

**Comment:**

The paper propose a universal technique that enables weak supervision over any label type while still offering desirable properties, including practical flexibility, computational efficiency, and theoretical guarantees.

Over the course of the rebuttal, the authors have made a substantial overhaul on writing and experimentation. The universality claims are now better supported by bounds, and experiments cover comparison to snorkel, majority vote and supervised learning, on multiple applications. The authors are encouraged to move the related work section to the main body of the paper. The authors should also clarify to what extent the contributions they make pertain to Snorkel as opposed to weak supervision more generally. This may require revisiting both the introduction as well as perhaps the title.